# LawFlow: Collecting and Simulating Lawyers' Thought Processes on Business Formation Case Studies

**Debarati Das[1], Khanh Chi Le[1]***, **Ritik Sachin Parkar[1]***, **Karin De Langis[1],**
**Brendan Madson[2], Chad M. Berryman[2], Robin M. Willis[2], Daniel H. Moses[2],**
**Brett McDonnell[2], Daniel Schwarcz[2], Dongyeop Kang[1]**
[1]Computer Science and Engineering, University of Minnesota
[2]Law School, University of Minnesota
{das00015,bhm,schwarcz,dongyeop}@umn.edu

## Abstract

Legal practitioners, particularly those early in their careers, face complex, high-stakes tasks that require adaptive, context-sensitive reasoning. While AI holds promise in supporting legal work, current datasets and models are narrowly focused on isolated subtasks and fail to capture the end-to-end decision-making required in real-world practice. To address this gap, we introduce LawFlow, a dataset of complete end-to-end legal workflows collected from trained law students, grounded in real-world business entity formation scenarios. Unlike prior datasets focused on input-output pairs or linear chains of thought, LawFlow captures dynamic, modular, and iterative reasoning processes that reflect the ambiguity, revision, and client-adaptive strategies of legal practice. Using LawFlow, we compare human and LLM-generated workflows, revealing systematic differences in structure, reasoning flexibility, and plan execution. Human workflows tend to be modular and adaptive, while LLM workflows are more sequential, exhaustive, and less sensitive to downstream implications. Our findings also suggest that legal professionals prefer AI to carry out supportive roles, such as brainstorming, identifying blind spots, and surfacing alternatives, rather than executing complex workflows end-to-end. Our results highlight both the current limitations of LLMs in supporting complex legal workflows and opportunities for developing more collaborative, reasoning-aware legal AI systems. All data and code are available on our project page.[1]

## 1 Introduction

Legal professionals, especially those early in their careers, face growing pressure to handle increasingly complex tasks - from navigating regulatory compliance to conducting detailed contract negotiations - while managing limited time and resources. Despite the promise of AI in legal domains (Schwarcz et al., 2025; Nielsen et al., 2024), current legal AI solutions remain narrowly scoped. They are often designed for isolated tasks such as contract review or legal research (Li et al., 2024b; Narendra et al., 2024), failing to reflect the broader arc of legal work. To realize AI's potential in legal practice, we argue for a shift in how legal reasoning is represented: not as isolated inputs and outputs, but as full, end-to-end decision processes.

Capturing legal workflows poses distinct challenges not seen in traditional automation domains. Existing automation and process modeling datasets emphasize deterministic sequences optimized for efficiency. Legal workflows, by contrast, demand interpretive, context-aware reasoning. Practitioners must frequently navigate uncertainty, integrate new information, revise decisions, and balance competing constraints. To build AI systems that support such work, we propose moving toward modeling not only final outcomes, but also the *processes*, or the "chain-of-decisions" that shape them.

---

*Equal Contribution
[1]https://minnesotanlp.github.io/LawFlow-website/

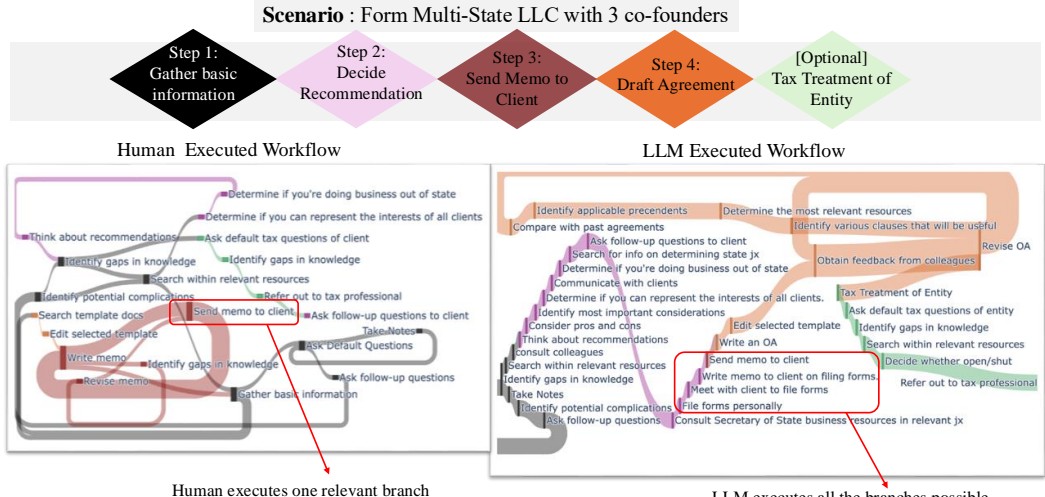

Figure 1: **Comparison of Human (law student) and LLM execution workflows for the same drafting task involving entity formation.** Here, *execution* refers to the selection of the next step in a reasoning process, not task completion, but the model's decision about what to do next based on context and prior actions. While the human execution is adaptive and iterative, the LLM follows a rigid, linear path, often executing all branches rather than selecting one relevant branch from the task plan. Points of divergence highlight differences in strategy, task decomposition, and assumption handling, revealing key limitations in LLM reasoning and opportunities for complementary human-AI collaboration.

Transactional law (Goforth, 2017; Feldman & Nimmer, 1999), and in particular business entity formation for small commercial ventures, offers a structured yet complex setting in which to study legal reasoning. Drafting an agreement involves eliciting incomplete client information, interpreting legal structures, and adapting language to evolving client goals and regulatory constraints (Goforth, 2017; Feldman & Nimmer, 1999). While prior datasets focus on static clauses or discrete legal tasks, they do not capture how legal practitioners navigate such workflows from start to finish. Legal reasoning in practice is dynamic and non-linear. It involves iterative refinement, contextual judgment, and responsiveness to client needs - patterns that challenge the rigid, task-specific strategies of current AI systems. To address this, we argue that AI must engage with the reasoning processes that underlie legal decision-making, not just the surface-level outputs. Traditional chain-of-thought (CoT) reasoning datasets (Wei et al., 2022; Kim et al., 2023) are insufficient for this goal. They typically model reasoning as linear, single-turn problems with a single correct answer. Legal workflows, by contrast, require what we term "chain-of-decisions reasoning": multi-turn, context-aware processes that involve ambiguity, revision, and multiple plausible paths forward (Wang et al., 2024b). This richer mode of reasoning better reflects the realities of legal practice, where flexibility and adaptation are essential.

To support this shift, we introduce LawFlow, a dataset that captures full end-to-end legal workflows from trained law students working through realistic business formation scenarios. By tracing the full decision arc - from client intake to operating agreement drafting - LawFlow enables fine-grained analysis of how human and LLM-generated workflows differ in structure, execution, and adaptability.

The chain-of-decisions reasoning captured in LawFlow allows for comparisons between human and LLM-generated workflows. For example, we can identify *points of divergence*, where strategies, task decomposition, or assumptions differ (highlighted in Figure 1). These divergences reveal limitations in current LLMs such as difficulties handling ambiguity, anticipating downstream implications, or revising incomplete plans which are failures that are not visible through traditional CoT evaluation alone. However, emerging reasoning-focused models (Guo et al., 2025; Jaech et al., 2024) show promise in addressing some of these issues, particularly in maintaining coherence across steps and accounting for long-range dependencies. Studying these differences also raises the possibility of mutual improvement:

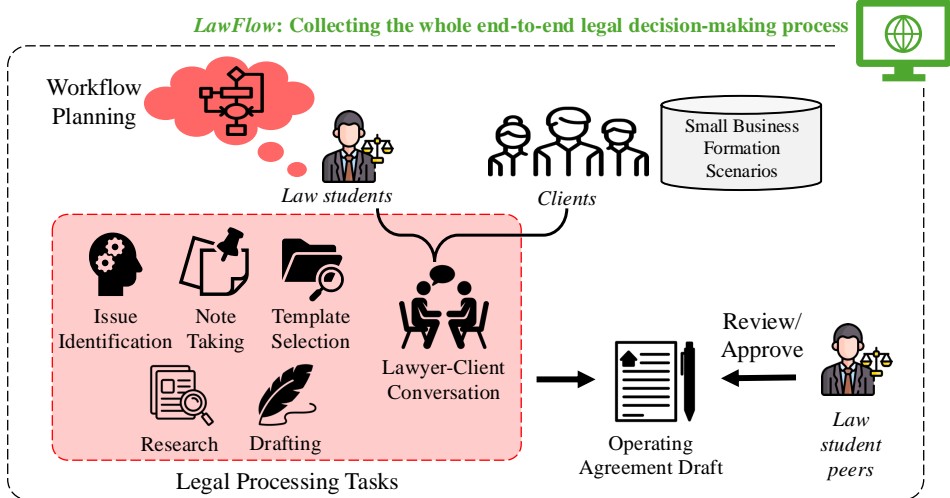

Figure 2: **Overview of the `LawFlow` dataset creation**, which captures novice law practitioner decision-making in drafting operating agreements for small business formation. Starting with realistic business formation scenarios, law students and clients simulate authentic client-information elicitation sessions. This is followed by sub-processes such as issue identification, note-taking, legal research, template selection, and drafting, with every human decision and corresponding reasoning recorded by the `LawFlow` data collection tool. The final operating agreements and memos are then reviewed and verified by other law students, emulating the oversight of senior partners in a law firm.

how might AI learn from human flexibility, and how might humans benefit from AI's structure or recall?

`LawFlow` supports research on key questions at the intersection of legal reasoning and LLMs:

1. How do human and LLM-generated legal workflows differ in structure, execution, and adherence to task plans?
2. Can we model diverse legal workflows and flag unlikely or anomalous steps, without presuming a single correct path?
3. What key decision points shape legal workflows, and how might AI systems support users at these junctures?

**Main Findings.** The fine-grained procedural data captured by `LawFlow` provides insight into these questions. Specifically, we find that human workflows tend to be more modular and adaptive, with subtasks often revisited or reordered based on new information. In contrast, LLM-generated workflows typically follow a rigid, linear sequence, with limited revision or backtracking. These findings reinforce the need for AI systems that support flexible, context-sensitive reasoning.

## 2 Background

**Lack of focus on legal workflows**. Current legal AI progress largely targets narrow tasks like legal classification (Lee, 2023), judgment prediction (Sesodia et al., 2025; Gray et al., 2024), contract comparison (Narendra et al., 2024), and legal research (Li et al., 2024b; 2023). These tasks are easier to formalize, enabling annotated datasets for evaluation and fine-tuning. While such datasets accelerate subtasks, they fail to capture the complex, multistep nature of real legal practice. LLMs excel at classification but struggle with application-oriented challenges requiring procedural reasoning and context (Guha et al., 2023; Li et al., 2024a). Legal workflows involve argument building, filings, interpreting metadata and are iterative, human-centric, and involve ethical trade-offs which benchmarks cannot capture (Choi et al., 2024). As AI saturates existing tests (Lawyer, 2024), these benchmarks inadequately reflect real utility. Though task-oriented AI aids legal work (Schwarcz et al., 2025), integrating

capabilities into broader workflows is critical. *Bridging this gap requires modeling realistic dynamic, structured, human-in-the-loop legal processes.*

**Inability of AI to successfully capture legal workflows** Unlike enterprise robotic process automation systems (Wewerka & Reichert, 2020; Ferreira et al., 2020), which handle repetitive tasks via rules, legal workflows demand domain-specific, adaptive reasoning amid ambiguity and negotiation (Herm et al., 2020). Autonomous process automation approaches using LLMs show promise in domains like travel planning (Ye et al., 2023; Fan et al., 2024; Xie et al., 2024; Wornow et al., 2024), yet even GPT-4 struggles with orchestrating complex, multi-step processes requiring procedural nuance (Zeng et al., 2023). *These models oversimplify workflows, failing to maintain coherence or interpretive reasoning, highlighting AI's limits in managing expert-driven workflows.*

**From Chain-of-Thought to Chain-of-Decisions: A Shift in Reasoning Paradigms** CoT datasets (Wei et al., 2022; Kim et al., 2023) train models for linear, single-task reasoning (Wang et al., 2024a; Guo et al., 2025), but real-world domains like law require dynamic, multi-turn "chains of decisions" shaped by evolving contexts, client goals, and strategic trade-offs. *Such workflows involve branching, revision, and subjectivity, which CoT-trained methods cannot capture* (Chen et al., 2024; Wang et al., 2024b). Inspired by studies logging scholars' writing processes (Wang et al., 2025; Koo et al., 2023), we argue that legal workflows demand explicit modeling of high-level reasoning and decision-making beyond document drafting.

## 3   LawFlow Dataset Creation

**Human Task Plan Construction**. We conducted in-depth interviews with senior law faculty and law students to understand how lawyers handle business entity formation towards drafting an Operating Agreement and modeled the entire end-to-end workflow. The resulting expert-informed task plan (Appendix Figure 5) captures this process across three levels of granularity: **inter-subtask** (the major workflow stages), **intra-subtask** (the interconnected steps within each stage), and **node-level** (individual actions annotated by cognitive modality). These modalities include introspective (individual reasoning), interactive (communication with clients or colleagues), and observable (engagement with tools or resources). The workflow reflects real-world legal practice, spanning information gathering, strategic decision-making, client communication, and iterative document drafting while highlighting how law students coordinate tools, knowledge, and judgment throughout. The main subtasks involved in this task plan are :

1. *Information Gathering:* The lawyer elicits basic client information, identifies legal or factual complexities, takes notes, and formulates follow-up questions. This subtask involves active interaction and may require consulting external resources for jurisdiction-specific issues.
2. *Deciding a Recommendation:* Based on gathered information, the lawyer assesses legal and strategic dimensions, determines jurisdictional applicability, and decides whether to proceed with a recommendation. They also evaluate their ability to represent the client and may schedule a follow-up.
3. *Drafting and Sending a Memo:* The lawyer formalizes their analysis in a client-facing memo outlining the proposed structure and areas of uncertainty or risk, revising as needed based on feedback or new information.
4. *Drafting the Operating Agreement (OA):* Using templates or precedent documents, the lawyer drafts the OA, tailoring clauses to client needs through iterative analysis, peer feedback, and revision.
5. *Assessing Tax Treatment [Optional]:* When tax issues arise, the lawyer gathers information, researches, and may refer the client to a specialist.

**Simulation of Legal Workflow Scenarios**. To contextualize our task decomposition and analyze entity formation workflows, we focus on high-performing third-year law students with business law experience - representative of novice practitioners whom educational and assistive AI systems are designed to support through scaffolded reasoning and guided task execution, rather than to replace as experts. One of the participating law students created a set of realistic seed scenarios which were based on anonymized real-world cases

from an affiliated legal clinic. Each scenario poses a unique small business formation challenge, crafted to surface varying legal considerations, client needs, and decision-making points. More details about these scenarios are in Appendix A.1.2. To simulate realistic legal interactions, we conducted structured roleplays where law students play-acted as lawyers advising clients. Computer science students were assigned the role of clients, enabling rich, interactive simulations of the legal process. During these roleplays, law students used the LawFlow Tool (described below) to document their reasoning, decisions, and actions. Each action was tagged with the corresponding subtask from our expert-informed legal task plan, allowing us to trace how legal workflows unfold in real-time.

**LawFlow Data Collection Tool**. To support data collection during legal roleplays, we developed a web-based application designed to reflect key elements of a law student's workflow during entity formation. Appendix Figure 8 showcases the different views in this web application. The tool includes several core features:

- *Subtask Annotation*: law students or the users of this tool, can tag each action with a relevant node from the expert task plan. Subtasks are non-sequential and optional, allowing for naturalistic variation in how people approach the task. Users can also annotate their decision-making rationale alongside each tag.
- *Note Taking*: A lightweight, persistent text field lets users take informal notes (e.g., client intake), with full keystroke logging for detailed analysis.
- *Template Library*: Users access curated Agreement templates, simulating real-world precedent use. Search activity is logged to capture document reuse.
- *Legal Search Interface*: Users can conduct open web searches and template-specific searches, simulating the use of proprietary legal research tools and allowing us to study how search behavior influences decision-making.
- *Document Editing*: A built-in word processor (using TinyMCE [2]) supports legal drafting. All keystrokes are logged, enabling reconstruction and analysis of the drafting process.

These features enable comprehensive capture of legal decision-making, tool usage, and workflow adaptation. Thus, the LawFlow Data Collection Tool allows us to observe what decisions are made and how law students navigate uncertainty, balance competing constraints, and structure their reasoning in practice. Appendix Figure 7 highlights the difference in human workflows for different scenarios.

**Dataset of Legal workflows**. The entity formation workflow typically produces two key artifacts: an *Agreement* and, optionally, a *client memo*. The Agreement is generated by selecting a template from a curated library and customizing it to fit the client's business structure, goals, and legal needs, through clause additions, removals, or edits informed by legal reasoning and client input. The client memo serves as a communication tool, explaining key decisions, legal obligations, and any open issues. Together, these documents reflect both the technical drafting and the student's interpretive reasoning.

Our collected dataset includes 10 finalized scenarios. While 20 were initially collected, the first 10 were excluded due to task plan revisions and RA acclimation to the tool. The final dataset covers diverse legal contexts (10 memos and 8 agreements) across multiple entity types like LLCs, non-profits etc). In 2 cases, the RA opted not to draft an agreement due to client representation conflicts or limited expertise, which are realistic outcomes reflecting legal ethics and complexity. Sessions lasted an average of 101 minutes. Appendix Tables 3 and 4 illustrate the complexity and diversity in our dataset. While our focus was on human and LLM workflows, we also evaluated output agreement quality through human and LLM reviews. To support this, experts compared LLM and human drafts for clarity, coverage, and issue-spotting. In some cases, LLMs were preferred or surfaced overlooked issues, showing potential as second-pass reviewers. These findings are detailed in the Appendix A.3.4.

## 4 Analysis and Discussion

This section presents our experimental framework and presents results for comparing how law students (humans) and LLMs plan and execute the full end-to-end legal workflows,

---

[2]https://github.com/tinymce/tinymce

instead of a single task execution. We focus on three main areas: (4.1) identifying structural and behavioral differences between human and AI workflows, (4.2) examining the diversity and monitoring of legitimate workflow paths, and (4.3) pinpointing critical decision moments where AI might assist.

**Construction of Plans and Execution Graphs.** To conduct our analysis, we collected structured task plans and execution traces from both law students and LLMs. These plans and execution traces can be visualized as graphs with nodes representing the subtasks and edges representing the transitions between them. Senior law faculty and law students helped develop an "near-ideal" *Human Task Plan* (Section 3), while law students participated in scenario-based roleplays using the LawFlow data collection tool, which logged their actions and decisions.

From these logs, we constructed a *Human Execution Graph*, Deviations from the Human Plan highlight reordering, omission, or improvisation. As SOTA reasoning models are known to be trained to be effective for multi-step planning, we prompted reasoning-focused LLMs like GPT-O1 and Deepseek-R1 (Jaech et al., 2024; Guo et al., 2025) - each independently and with identical scenario inputs - to generate two outputs: (1) *LLM Task Plan* – A high-level plan generated from multiple business scenarios. It aims to capture a broader strategy that covers various use cases, rather than a single scenario. (2) *LLM Execution Graph* – A trace of decisions about which task to perform next in a given scenario. Here, *execution* refers not to carrying out a task, but to the model's decision about what the next step should be, given the current context, prior actions, and scenario-specific details. Each step simulates a legal reasoning move-either a hypothetical client interaction or internal deliberation, followed by a next-step decision. The executed steps are then collected to form the *LLM Execution Graph*. Prompts and outputs for these steps are detailed in A.1.4. This representation differs from traditional chain-of-thought (CoT) reasoning, which typically involves linear, forward-only justifications toward a known answer. Instead, our graphs capture chain-of-decisions reasoning: multi-turn, context-aware navigation through a branching task space, often involving revision, ambiguity, and multiple plausible paths.

## 4.1 How do human and LLM-generated legal workflows differ in structure, execution, and adherence to plans?

We investigate how legal workflows differ when generated and executed by law students versus LLMs, focusing on three dimensions: (1) the structural characteristics of task plans, (2) adherence to those plans during execution, and (3) behavioral patterns during task performance.

**Task Plan Structure.** We want to understand the differences between humans and LLMs in planning out legal workflows and what those differences look like structurally. In service of that, we explore the modularity and branching of their task plans via the following graph-based metrics: *Average tree depth* (the number of hierarchical layers in the task plan), *Number of top-level subtasks* (reflecting the initial modular structuring), *Average branches per subtask* (capturing the extent of decomposition), and *Average nodes per subtask* (measuring the level of internal detail).

Our results indicate that human task plans are modular and hierarchical, often beginning with a few high-level subtasks that are incrementally decomposed into finer-grained actions (Figure 3). In contrast, LLM-generated plans are flatter and more linear, typically enumerating long lists of actions without deeper structural organization. Table (a) in Figure 3 shows that human plans exhibit greater tree depth (3.6 vs. 1.0) and more nodes per subtask (8.8 vs. 3.8), indicating deeper and more layered decompositions. LLM plans, on the other hand, contain more top-level subtasks (10 vs. 5), reflecting a breadth-first approach with less internal depth. Figure 3(b) shows that law students plan by starting with a small set of high-level goals and anticipate that they may need to decompose them by identifying knowledge gaps or consulting with clients. By contrast, AI's task plan lays out a flat, exhaustive task list with little hierarchical structure and more focus on the nitty-gritty details. These patterns reinforce a key distinction: *human plans emphasize modularity and depth, while*

| Graph Metrics | Human Plan | LLM Plan |
|---|---|---|
| Tree Depth | 3.6 ± 0.5 ↑ | 1.0 ± 0.0 |
| Number of Subtasks | 5 ↓ | 10 |
| Branches/Subtask | 4.8 ± 2.3 ↑ | 3.8 ± 0.4 |
| Nodes/Subtask | 8.8 ± 2.6 ↑ | 3.8 ± 0.4 |

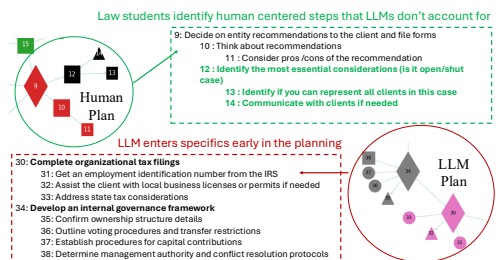

**(a)** Humans plan hierarchically, starting with a few high-level subtasks (number of subtasks) and expanding them into deeper, more detailed structures (tree depth and branches/subtask). In contrast, LLMs generate flatter, sequential plans with more top-level subtasks but less internal organization and modularity.

**(b)** The figure shows LLM task plan nodes colored by subtask, and the human task plan is mapped to the LLM task plan based on semantic similarity of subtasks. We can see multiple LLM subtasks correspond to the same human subtask. Black nodes indicate nodes which remain unmapped in the human plan (indicating that there are some human tasks LLMs do not account for).

Figure 3: **Difference between Human and LLM Plans.** Humans plan hierarchically with few high-level subtasks and deeper trees. LLMs use flatter structures with more top-level subtasks but lower internal organization.

*LLM plans prioritize surface-level breadth.* The expanded version of both plans are provided in the Appendix Figure 12.

**Task Plan Adherence.** We also investigate how closely each agent, human or LLM, follows its own plan during execution. This helps us understand not just how workflows are formulated, but how faithfully they are carried out in practice. To quantify adherence, we use two complementary metrics: *Node Execution Rate* (the proportion of planned actions that were actually executed) and *Levenshtein Distance* (the textual similarity between the linearized plan graph and execution graph; lower values indicate closer alignment).

Table 1a compares how closely each agent follows its own plan. LLMs tend to stick more closely to their original task plans, with a higher node execution rate (0.78 vs. 0.54) and lower Levenshtein distance between planned and executed sequences. This suggests a more rigid and deterministic execution style. Human executions show greater deviation, consistent with the intuition of a more adaptive workflow that involves reordering, skipping, or refining tasks in response to real-time judgment.

**Structure-based Execution Behavior.** Understanding how agents (human or LLM) carry out legal work is just as important as how they plan it. While structural differences in task plans reveal how humans and LLMs conceptualize workflows, their execution behavior sheds light on the reasoning strategies they employ in practice-whether they follow a fixed linear path, adapt in real-time, or select steps dynamically. To investigate this, we compute the following metrics to capture these execution dynamics: *Number of cycles* (how often tasks are revisited), *Average visit count per node* (frequency of re-engagement with individual steps), *Subtask group transition count* (how often execution shifts between subtask categories), and *Node execution rate* (repeated here to connect execution behavior to plan coverage).

When executing the same expert-defined plan, humans exhibit highly dynamic behavior. As shown in Table 1b, they revisit subtasks frequently (17.5 cycles), re-engage individual steps (2.3 average visits), and shift between subtask categories (13.2 transitions), suggesting non-linear, adaptive execution. LLMs, by contrast, show no cycles, visit each node only once (1.0), and exhibit fewer transitions between subtask categories. Their node execution rate is also higher (0.8 vs. 0.5), reflecting more exhaustive coverage of the original plan, but with less variation in execution flow.

| Metrics | Human Exec (Human Plan) | LLM Exec (LLM Plan) |
|---|---|---|
| Node exec rate | $0.53 \pm 0.29$ | $0.78 \pm 0.14 \uparrow$ |
| Levenshtein dist. | $49.30 \pm 19.59$ | $10.70 \pm 6.53$ (before) $37.90 \pm 0.74$ (after) $\downarrow$ |

(a) **Adherence to Task Plan**. LLMs exhibit higher adherence to their own task plans than humans, with higher node execution rates and lower textual deviation. We also align the LLM-execution with the human execution, thus (before) shows the scores pre-mapping and (after) shows the scores post-mapping.

| Metrics | Human Exec (Human Plan) | LLM Exec (Human Plan) |
|---|---|---|
| # of cycles | $13.40 \pm 16.24 \uparrow$ | $0.40 \pm 0.70$ |
| Visit count | $2.18 \pm 0.65 \uparrow$ | $1.04 \pm 0.05$ |
| Sub-task trans. | $12.80 \pm 6.12 \uparrow$ | $5.30 \pm 0.48$ |
| Node exec rate | $0.53 \pm 0.29$ | $0.77 \pm 0.04 \uparrow$ |

(b) **Execution patterns**. Humans execute plans non-linearly with cycles and frequent transitions; LLMs execute linearly with fewer deviations.

Table 1: Comparison between Human and LLM execution of task plans, across human- and LLM-generated plans.

**Granular Assessment**. To further characterize these behaviors, we analyze execution at three levels described in Appendix Figure 5:

- *Node Level* - We examine how specific tasks are performed, focusing on "selecting a business entity type", which has significant downstream implications for the final agreement. We measure the match rate, or how often LLM choices align with those made by humans.
- *Intra-subtask Level* - This level assesses how thoroughly agents complete the steps within each subtask. We use a coverage metric to check whether all expected actions (as defined in the human plan) are carried out. We also calculate the Levenshtein distance between each agent's execution and the human plan to measure how much their workflow diverges through edits.
- *Inter-subtask Level* - We evaluate how agents transition between subtasks. Using the Longest Common Subsequence (LCS), we analyze the presence of cycles or deviations in task order. We also compute the total number of edits (e.g., insertions) across scenarios to determine which agent alters the planned sequence more frequently.

We see in Appendix Table 5 that, at the node level, LLMs matched human entity selection 60–70% of the time post-client elicitation, which may affect the final agreement. At the intra-subtask level, humans prioritized early-stage actions (e.g., client intake) and often skipped lower-yield steps, while LLMs distributed effort more evenly, especially emphasizing operating agreements and tax steps, even when tax was minimally relevant. Humans showed more context-sensitive and iterative behavior, with greater reordering (higher Levenshtein distance). At the inter-subtask level, humans frequently looped back or revised steps, whereas LLMs followed rigid, sequential workflows.

## 4.2 Can we model diverse legal workflows and flag unlikely steps?

Legal workflows are highly variable, shaped by context, judgment, and personal strategy. Even among trained law students, the same scenario can yield multiple valid paths. This diversity reflects the flexibility of legal reasoning, not noise. To support AI systems in this space, we analyze how workflows vary and introduce a lightweight "workflow monitor" tool to flag steps that diverge from common human patterns, prompting reflection without enforcing correctness.

**Characterizing Workflow Diversity.** We begin by analyzing workflows under two conditions: (1) the same law student executing different legal scenarios, and (2) different law students handling scenarios of similar complexity. Figure 4 shows examples of subtask connectivity in these cases. The first two panels demonstrate how the same individual varies their strategy across contexts, while the latter two show *different individuals approaching simi-*

*lar scenarios in distinct but valid ways*. These observations highlight the flexible, interpretive nature of legal work and highlight the limits of assuming a single canonical workflow. They also motivate the need for datasets and models that accommodate this variation in reasoning style.

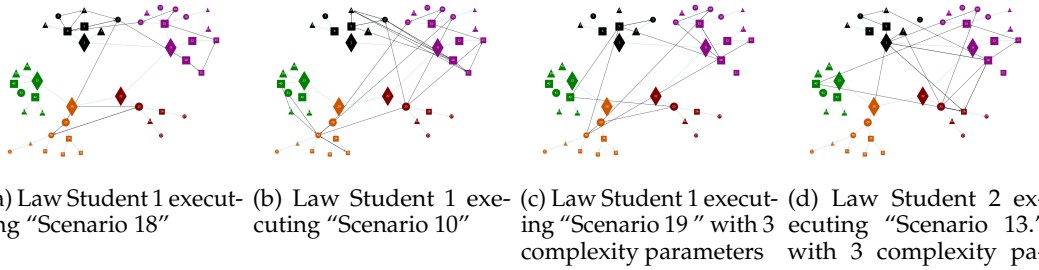

(a) Law Student 1 executing "Scenario 18"

(b) Law Student 1 executing "Scenario 10"

(c) Law Student 1 executing "Scenario 19" with 3 complexity parameters

(d) Law Student 2 executing "Scenario 13." with 3 complexity parameters

Figure 4: **All workflows matter**. Comparison of subtask connectivity across humans and tasks. The first two figures show the same human performing two different tasks. The last two figures show different humans performing tasks of similar complexity. Despite the variations, each workflow is a valid approach to completing the task. Descriptions of these scenarios are present in A.1.2.

**Modeling Atypicality.** To identify steps that diverge from typical human reasoning patterns, we train a step-level workflow monitor using the LLaMA-3.2B-Instruct model (Grattafiori et al., 2024) via supervised fine-tuning on 1,295 steps from eight human-generated workflows, reserving two human (95 steps) and ten LLM scenarios (375 steps) for evaluation. At each step, the model predicts the most likely next action given prior context and assigns a perplexity score. We define a deviance score as the z-score of that step's perplexity relative to the distribution observed in human training data. Lower scores indicate alignment; higher scores suggest atypicality - not as errors, but as moments that may benefit from review. These Z-score bands are shown in Appendix Table 6. Rather than using a classifier, this perplexity-based approach enables soft anomaly detection of atypical steps, confirmed by 90% accuracy in a manual review of 40 evaluation steps (20 human, 20 LLM). Results (Appendix 7a) show that *67.8% of LLM-generated steps align closely with human behavior*, while others fall into moderate and high deviation bands. These deviations often reflect context-insensitive decisions or uncommon execution paths. Thus the workflow monitor provides a soft signal to support reflective execution. It helps surface unusual transitions in real-time and could be extended to DPO-style preference tuning with annotations. Additionally, beam search visualizations in Appendix Figure 14 show that human workflows tend to follow typical, well-aligned subtask sequences, while LLMs exhibit more scattered and atypical transitions.

### 4.3 What key decisions shape legal workflows, and how can AI help?

Some decisions in legal workflows disproportionately shape downstream outcomes. We refer to these as **meta-decision points**, points where the choice or action taken can reframe the task, influence later reasoning, or affect the final deliverable. Identifying these points helps us understand where workflow divergence is most likely and where AI assistance may offer the greatest value. To surface meta-decision points, we combine graph-based analysis with human judgment. We compute two metrics across nodes in the task diagram: (1) *Betweenness Centrality* – how structurally influential a node is in connecting the workflow and (2) *Time Spent* – average human engagement time per node, signaling complexity or perceived importance We also collect input from law students about which decisions they see as most critical. Results (Appendix Figure 16b) show that early-stage actions like note-taking and client questioning are central and time-intensive, especially in Subtask 1. Law students identified these moments as ideal for AI support in surfacing gaps and refining strategy. In contrast, they preferred AI to take a review role during later drafting stages (Subtask 4), highlighting key opportunities for context-sensitive, high-impact AI interventions.

# 5 Conclusion

This paper introduces `LawFlow`, the first empirical dataset capturing complete legal decision-making workflows by trained law students engaged in realistic drafting tasks. Unlike prior datasets focused on isolated subtasks, `LawFlow` models how legal reasoning unfolds in the face of uncertainty, incomplete information, and evolving client context.

We compare human and LLM reasoning using two methods: logging human role-plays and prompting LLMs for next-step decisions. While this enables direct comparisons, observed LLM linearity may reflect prompt design, and black-box inference limits visibility into internal reasoning. Future work should explore alternative prompting or think-aloud protocols to better align human and model analysis.

Taken together, `LawFlow` serves as:

1. a **divergence lens** to examine how legal reasoning pathways vary across agents,
2. a **test bed** for lightweight workflow monitors that flag atypical steps without enforcing a fixed solution, and
3. a **seed corpus** for building collaborative legal assistants that visualize progress, surface alternatives, and intervene at key decision points.

Extending `LawFlow` with richer traces and broader legal scenarios will deepen its value. Even in its current form, it shows that modeling decision processes - not just outcomes - is key to developing AI systems that truly augment the reflective, context-sensitive nature of real-world legal practice. Though focused on business formation, LawFlow captures complex reasoning patterns - like iterative refinement and contextual adaptation - that generalize across legal domains. We also outline design suggestions in Appendix A.4, grounded in these findings, to inform the development of such systems.

# 6 Limitations

Creating the `LawFlow` dataset involved several challenges. Capturing end-to-end legal workflows is inherently difficult, likely due to the high human effort, expert oversight, and coordination required. Our approach relied on a structured human task plan to guide data collection. While necessary for consistent annotation, this framework may have influenced participants to follow the given structure, reducing the natural variability typical of legal practice. Additionally, the entity formation scenarios are intentionally simple (Appendix A.1.2) to isolate reasoning patterns, though they may underrepresent real-world complexity. Future versions will incorporate more nuanced, expert-verified cases.

Modeling authentic legal behavior also meant accommodating the unpredictability of human actions. Participants sometimes used external tools like Westlaw or Google - actions that, while noisy for structured logging, reflect actual practice. Rather than treat these as errors, we view them as important signals of practitioner behavior and may explicitly log them in future versions.

Annotation and evaluation introduced further complexity. Standard metrics like task time are confounded by learning effects, so we advocate for behavior-based measures like decision consistency. Divergence between human and AI outputs is expected given the subjectivity of legal reasoning and offers insight into alignment and where automated systems fall short of expert judgment.

# Acknowledgments

This work was mainly supported by the Agent Benchmark Grant from the Open Philanthropy. We also thank Minnesota NLP group members for providing us with valuable feedback and comments on the initial draft. We are grateful to Kiri Somermeyer at UMN Law's Business Law and Entrepreneurship Clinic for sharing business formation scenarios and offering valuable feedback.

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

# A Appendix

## A.1 More about Data Collection

### A.1.1 Expert-Informed Task Plan

We conducted interviews with senior law faculty and students to model the end-to-end workflow of business entity formation leading to an Operating Agreement. The expert-informed task plan in Figure 5 spans three levels: major workflow stages (inter-subtask), steps within each stage (intra-subtask), and individual actions tagged by cognitive modality - introspective, interactive, and observable. Key subtasks include gathering client information, forming legal recommendations, drafting memos, composing Operating Agreements, and optionally assessing tax implications (Figure 6). The workflow reflects real-world legal practice, emphasizing how novice lawyers coordinate reasoning, research, tools, and communication to navigate complex, evolving client needs in a structured yet adaptive process.

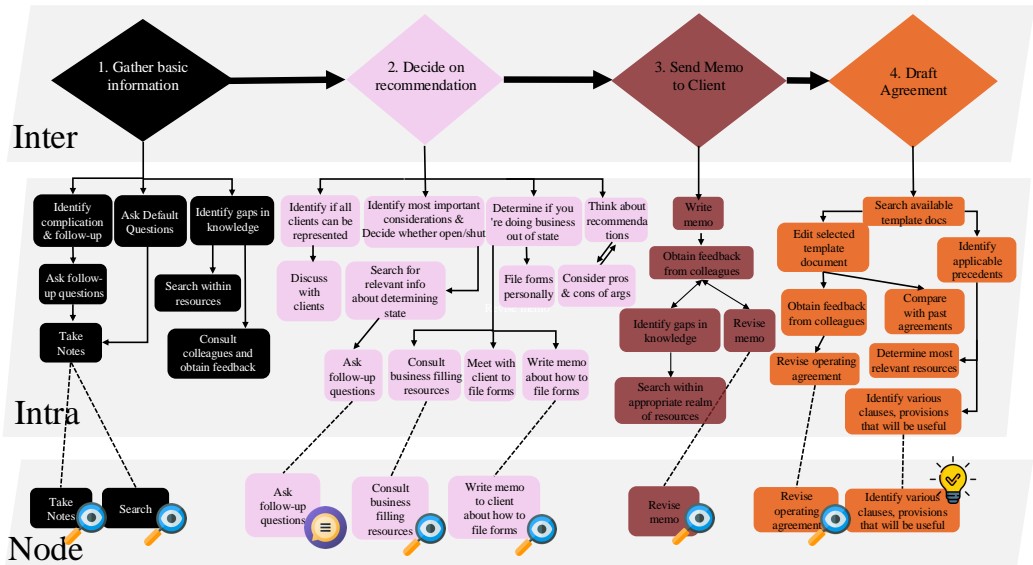

Figure 5: **Expert-informed task plan for business entity formation.** This diagram illustrates the multi-level structure of the legal workflow behind drafting an Operating Agreement. The inter-subtask level (top row) outlines the major stages of the process, from initial client intake to final document preparation. The intra-subtask level (middle row) decomposes each stage into finer-grained tasks and shows their interconnections. The node-level (bottom row) represents individual actions, annotated by cognitive modality: introspective (internal legal reasoning), interactive (client or colleague communication), and observable (use of external tools and resources). Thus, the task diagram captures the complexity, adaptivity, and tool-mediated nature of real-world legal reasoning.

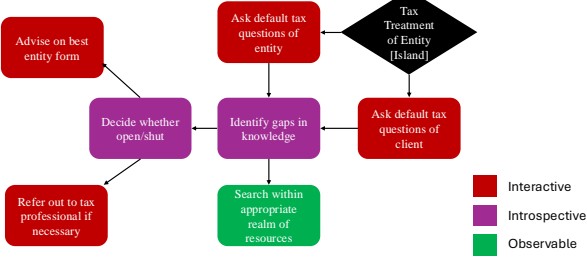

Figure 6: **Subtasks involved in dealing with Tax considerations** (considered mostly optional in our scenarios)

### A.1.2 Scenarios informing Data Collection

Here, we present some of the entity formation scenario's we have mentioned in our analysis in this paper. In addition to subtask tagging, each scenario is annotated with two metadata sets: *complexity* and *nuance* parameters, which are described in Table 2. These annotations allow us to capture factors that influence the legal task's overall structure and difficulty and the role-play's interactive dynamics. *Complexity parameters* identify structural features of a scenario that are likely to impact the scope and duration of the legal work and the outputs produced (e.g., operating agreement, client memo). These parameters influence how the lawyer navigates the workflow and which subtasks are emphasized, revisited, or escalated. *Nuance parameters*, by contrast, shape the interactivity and dynamism of the roleplay itself. These factors make the simulation more realistic and help model the types

of human-centered challenges lawyers regularly face. By systematically varying these parameters across scenarios, we are able to study how different contextual and interpersonal factors shape legal task execution. This, in turn, supports a more robust analysis of legal workflows, the adaptability of experts, and the design of systems that can assist with or simulate professional legal reasoning in diverse settings.

Table 3 summarizes the diversity and complexity of the scenarios in our dataset. Table 4 provides statistics on the types of business entities covered in our dataset and the lengths of the outputs and tool usage associated with each.

| Complexity Parameters | Nuance Parameters |
|---|---|
| <ul><li>Whether a client memo is needed</li><li>Whether separate lawyers are required for conflict management</li><li>Whether routing to a different lawyer is necessary due to lack of expertise</li><li>Number of individuals involved in the business</li><li>Nature of the business (e.g., regulated industry, high-risk sectors)</li><li>Business expansion plans</li><li>Number of investment sources</li><li>Diverging interests among founders or stakeholders</li></ul> | <ul><li>Need for follow-up with the client after the initial information-gathering session</li><li>Need for follow-up after delivering the first version of the operating agreement</li><li>Whether the client is overprepared (e.g., brings documents, has done prior research)</li><li>Whether the client is underprepared (e.g., lacks familiarity with legal or business concepts)</li><li>Whether multiple client sessions are needed to arrive at an entity choice decision (e.g., due to complex funding structures or high-stakes investments)</li></ul> |

Table 2: Parameters for **Complexity and Nuance** in Legal Consultations

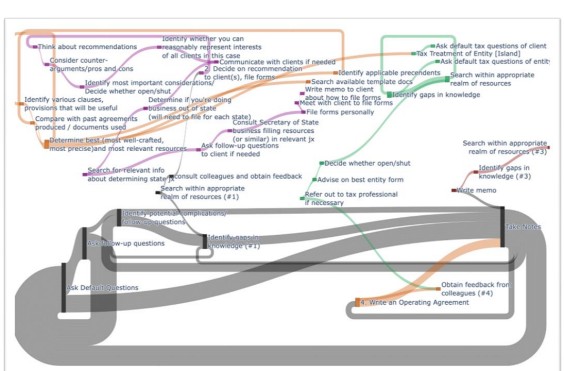
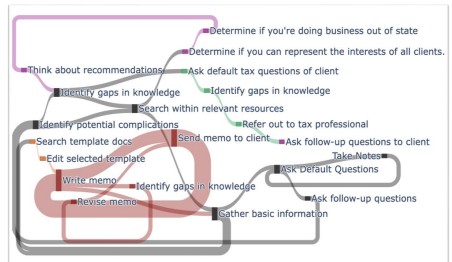

(a) **Human execution workflow for Scenario 1**: A and B are avid anglers aiming to promote fishing in our area by hosting free educational workshops for children and adults. They plan to form a business to manage gear purchases, raffles, and possible future funding, but want to avoid complex formalities and do not intend to make a profit. They are interested in understanding what kind of startup agreement they need.

(b) **Human execution workflow for Scenario 2**: A and B have created the chemical composition of an organic, temperature-regulating soil. A and B need additional funding to produce an amount large enough to be tested on a greater scale & form a viable business. A and B's friend C is willing to invest 2M for a 30% ownership interest, which A and B agree to. If all goes well, the three want to sell the soil's chemical composition to a biotech company.

Figure 7: **Comparing different human workflows**: In Scenario 1 where the client was uncertain about next steps, the workflow became more exploratory and recursive, with the student circling back to earlier tasks to clarify gaps. In contrast, the workflow was more streamlined in the more straightforward scenario 2, focusing on next steps and formalizing recommendations.

Some scenarios mentioned in the main paper, along with their descriptions are provided below. Figure 7 highlights the difference in human workflows for different scenarios.

**scenario 10**:

Title: Three-Person Coffee Truck with Entity Formation Contingency Context: Three friends, A, B, and C, want to form a coffee shop out of a food trailer that A purchased while the three were college roommates. They had informally used the trailer to sell coffee products on campus, but would like to see if they could form a real business. The key to their popularity has been A's importation of Hawaiian coffee beans, which help the trio brew delicious coffee drinks but has proven to be an expensive business decision. If they form an entity, they might want to see whether they could have investors supply some extra money to help them cover the cost of importing the beans. They would plan on giving the investors some kind of profit interest in the business in exchange for their help, but aren't too familiar with how that would work.

In addition, this venture has largely been A's project. B and C aren't quite as invested, as they have more time-consuming careers of their own. The group's other friends, D and E, however, have kept in touch with A and have expressed interest in joining the business in place of B and C. B and C might be open to this as well, but will need some time to think. A must also later decide if he wants anyone involved with the business to be able to come and go as they please, or if he wants a firmer commitment from anyone involved with the truck (either B and C or D and E) to devote more of their time and resources to it. Details: 1. Three-person 2. Sale of Goods 3. Personal Property 4. Entity Selection Complexity Tags: 1. Memo needed 2. Diverging Interests of Clients 8. Multiple Sessions Needed for Entity Choice Complexity: Medium

**scenario 13**:
Title: One Person Ice Cream Maker Context: E has recently developed a process for making premium homemade ice cream with locally-sourced ingredients. The result has been a hit: E's flavors are wide-ranging, unique, and high-quality. E has started by sharing the products with neighbors, who have suggested that he sell it for profit. E has not yet done so, but might be interested. He is wondering how to best proceed by beginning with local sales and going from there. Details: 1. One person 2. Inexperienced client Issues: 1. Formation 2. Branding 3. IP

**scenario 15**:
Title: Lake Bed and Breakfast Context: M and A want to operate a bed and breakfast on the lake where they keep a cabin – just outside a small town in the northern, rural part of their home state. They intend to run a relatively small operation: the building they have picked out is a four bedroom house that can accommodate up to ten people comfortably. It is in relatively good shape, but would benefit from a handful of basic renovations. M and A will cook breakfasts, provide housekeeping services, and lead hikes for guests in the surrounding area. They will also provide kayaks, stand up paddleboards, and jet skis for rent, all of which are personally owned by the couple. They have a sufficient (i.e. covering everything) liability waiver in place which guests agree to upon booking their stay. Per an informal agreement, M and A have also received some funding from the nearby town's chamber of commerce in exchange for recommending other local businesses to their guests. They're interested in forming a business to separate their personal assets from that of the business. In addition, they'd like to pass the business to their three children (ownership divided equally among them), and want to add them to the business now, each with partial ownership (M and A each holding a 25% ownership). They would also like a provision in any operating document to specify that, should anyone seek to sell their ownership interest, the remaining members must get the first offer or otherwise unanimously consent in writing to the transfer. Details: 1. Two Person 2. Services 3. Varying Liability 4. Transfer of Business Issues: 1. Entity Formation 2. Addition of Members 3. Restrictions on Interest Transfer Complexity Tags: 1. Memo needed 2. Diverging interests of clients

**scenario 17**:
Title: Fishing Education Context: A and B are each avid anglers and seek to grow the sport within their area, which is largely urban. They seek to teach fishing skills—ranging from basic to advanced—to both children and adults, through events held at parks, lakes, and rivers nearby. They envision weekly or monthly workshops where they educate participants on a particular skill—casting, knot tying, fish landing with a net, and more. Signup for the workshops is free. A and B encourage participants to bring their own fishing gear, but plan on contracting with a local sporting goods store to purchase bulk orders of tackle at a discounted rate. Because of that, they have decided to explore forming a business entity. Neither A nor B are particularly familiar with small business structure or ownership, and don't seek to profit from the business. They would like, however, to minimize any formalities associated with owning a business. In addition, they would like to raffle off fishing apparel at some of the workshops, and are ok using their own money to buy these items for the first few instances. If their workshops are popular and they consistently have high attendance, they would like to borrow money or obtain outside funding to keep this practice up. Overall, they want to learn more about the function of any documents used to start the business, and potentially review anything that's written for them before they officially get to work. Details: 1. Three Person 2. Low Liability 3. Informal Structure Complexity Tags: 3. Follow-up for Operating Agreement Review 7. Inexperienced Client

**scenario 18**:
Title: Shared Workspace and Kitchen Context: P is looking to bring together other businesses in his community by offering a space that functions as both a commercial kitchen and shared workspace, which can be leased by individuals or other entities. P has already formed an LLC for this purpose. The leases may be by the hour or by the day, and can be for the kitchen, spots in the workspace, or for the entire property. To minimize out-of-pocket expenses, P decides to obtain financing from a local bank for the necessary remodeling. In addition to the normal rentals, P has two other potential uses associated with the building. One is a regularly-occurring food truck fair that will feature local, newly-formed food trucks. It will take place on the building's property, in its parking lot. The second is a temporary restaurant featuring one of P's former business partners (B) who has become a chef. The restaurant will have the exclusive use of the space for ten weeks of the subsequent summer, and will be a collaborative effort between the two entities. P will be involved as well, as he and B had originally met while working together in another restaurant. Details: 1. One Person 2. Lease agreement needed Complexity Tags: 6. Well-prepared client

**scenario 19**:
Title: Refurbishing Hockey Equipment Context: L accepts used hockey equipment—generally, skates and sticks—and repairs it, sometimes to give back to those who donate it, and other times for resale, if the donating party has no further use for it. The work is mostly done in his home, and varies based on how much equipment he has at a given time. He's considering implementing the option of selling the resale

equipment to a local hockey equipment store, mostly depending on if he keeps getting a supply of broken sticks—the piece of equipment most common on the resale market. Though buyers at these stores are aware of the potential defects of a refurbished hockey stick, the stores selling them generally provide no warranty on them, effectively relieving them of liability. L is confident—and correct—that there are no huge liability risks associated with selling refurbished sticks, but might want some protection—as well as a more official entity—to do business with the stores, and the consumers he sells to on the side.

**scenario 20**:
A, B, and C have all worked together at a health clinic in the Twin Cities for the past 11 years. A and B are both physicians who have client relationships that would follow them from the current clinic to the one they are creating. Even though they will not be able to directly contact the clients from their current practice due to a non-compete agreement, they are well known and respected in the community and expect that many clients will follow them without proactive contact. C is an administrative professional who has practically run the current practice for the last 20 years and will be vital to the day-to-day operations of the new entity. C will be contributing all of the labor to running the new entity, including hiring support staff, scheduling appointments for the doctors, etc. The three have identified a suitable location for the new practice and may either purchase or lease it long-term depending on how the negotiations go. During the conversation, it should become clear that C expects an equal share of the new entity and is also worried about having decision-making power after formation. C does not have as much money to contribute as the doctors do and will be contributing mostly labor. A and B are somewhat dismissive of C's value and expect to be "running the show" themselves because of their extensive training as doctors, etc. They find C indispensable, but not necessarily worthy of equal ownership in the new entity. Tags associated : 1.Three-Person (just two clients needed for simulation) 2.Financial Considerations 3. Entity Growth Complexity Tags 1. Memo Needed Nuance Tags 4. Follow-up for Entity Choice

Table 3: **Diversity and Complexity of Scenarios** split by Category

| Category | Count | Domains | Complexity Summary |
|---|---|---|---|
| Nonprofit / Mission-Driven | 3 | Health; Education | Simple setup; compliance burden; no profit motive |
| Startups / Entrepreneurial | 3 | Food; Agriculture | Growth-focused; needs funding, IP setup, and lease structuring |
| Family / Lifestyle Business | 3 | Hospitality; Food; Sports Equipment | Often solo-run; asset or succession planning adds complexity |
| Professional Practices | 1 | Professional Services | High regulatory demands; ownership and ethical constraints |

Table 4: Calculated **Metrics for Business Entity** Formed Per Scenario

| Metrics | C-corp (1) | Nonprofit (1) | LLC (6) | No OA (2) | Total Avg (10) |
|---|---|---|---|---|---|
| Agreement length | 5815.00 | 5071.00 | 3704.67 ± 1158.00 | – | 4139.25 ± 1282.54 |
| Memo length | 262.00 | 514.00 | 380.17 ± 183.06 | 577.50 ± 207.18 | 421.20 ± 183.61 |
| Tools used | 43.00 | 35.00 | 77.67 ± 49.29 | 30.50 ± 20.51 | 60.50 ± 43.58 |

### A.1.3   Lawflow Data Collection Tool

The different views from the developed Lawflow Data Collection Tool are shown through Figure 8. Videos of the experience can also be seen on the project website.

### A.1.4   LLM Workflow Generation

**LLM Plan Graph Prompt**   The prompt we used for an AI generated task plan for the entity formation scenario is given in Figure 9. This is analogous to the human task plan.

**LLM workflow generation prompt**   The prompt we use to generate the AI workflow is given in Figure 10. This is analogous to the human workflow. A snippet of the output from this prompt is shown in Figure 11. The output of the LLM along with the next step suggested by it is fed as context to the LLM again to generate the details of the next step. To start the execution of a particular scenario by the LLM, we provide a prompt similar to Figure 10 but without the additional context as there are no prior steps to be input.

### A.2   Additional Analyses and Results

### A.3   Human vs LLM Plans

Figure 12 shows that multiple LLM subtasks often map to a single subtask in the human plan, indicating coarser granularity. Black nodes in the human plan represent tasks not

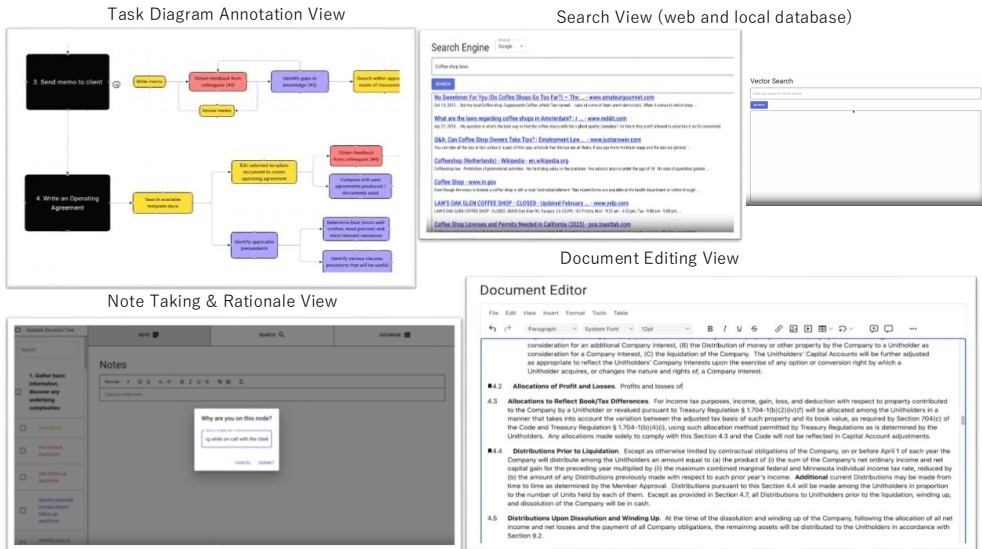

Figure 8: **Features of the LawFlow Collection Tool** include note taking, document editing, search as well as task diagram annotation.

```
You are an experienced lawyer specializing in entity formation at a reputable American
law firm. Your task is to create a structured, high-level reasoning plan outlining the
steps you would consistently follow for any business formation scenario, from initial
client consultation through the final drafting of the Operating Agreement.

To inform your thinking, consider these illustrative business scenarios:

<business_scenarios>
{BUSINESS_SCENARIOS}
</business_scenarios>

These scenarios are provided as examples to guide your reasoning. ....

1. Identify key components of entity formation
.
.

It's okay for this section to be quite long to ensure a thorough analysis.

After completing your analysis, ..

1. Clearly number the main tasks (e.g., "1. Task Name").
.
.
.

Example format [Only for demonstration purposes]:

1. Main Task One
    1.1. Sub-task A
    1.2. Sub-task B [Parallel/Concurrent]
        1.2.1. Sub-sub-task i
        1.2.2. Sub-sub-task ii
    1.3. Sub-task C
.
.

Important: Create your plan based on your professional....
```

Figure 9: LLM Task Plan Generation Prompt

captured in the LLM plan, for example "consulting colleagues and obtaining feedback" or "identifying gaps in knowledge" are common human steps which are not accounted for by the LLM. These patterns reinforce a key distinction: *human plans emphasize modularity and depth, while LLM plans prioritize surface-level breadth.*

```
You are an AI assistant tasked with generating realistic interactions related to small
business formation. Your goal is to create either a dialogue between a lawyer and clients
 or a description of a lawyer's actions, based on a specific step in a high-level plan
for business formation.

First, review the following information:

1. High-level plan for business formation:
<high_level_plan>
{task}
</high_level_plan>

2. Specific scenario for this business formation:
<scenario>
{scenario}
</scenario>

3. Previous steps and context [Note: The steps at the top are the earliest ones, and the
ones at the bottom are the latest ones.]:
<previous_steps>
{context}
</previous_steps>

4. Suggested step for this turn:
<suggested_step>
{suggested_step}
</suggested_step>

Your task is to generate either a conversation or a description of the lawyer's actions
based on the suggested step. Follow these instructions carefully:

1. Analyze the current step and plan the interaction:
<step_analysis>
.
.
</step_analysis>

2. Assess the lawyer's knowledge at this stage:
<knowledge_assessment>
.
.
</knowledge_assessment>

3. Generate the interaction:
If the step involves both lawyer and clients:
<conversation>
Lawyer: [Lawyer's dialogue]
Client: [Client's dialogue]
[Continue the conversation as needed]
</conversation>

If the step involves only the lawyer:
<lawyer_action>
[Describe the lawyer's actions, thought process, and any documents or research they might
 be working on]
</lawyer_action>

4. Suggest the next step:
<next_suggested_step>
[Specify the next step from the plan]
</next_suggested_step>

Remember:
- Ensure that your dialogue or action description is relevant and tailored to the given
business context.
.
.

### Answer
```

Figure 10: LLM Workflow generation prompt

| | | Metrics | Human exec | O1 exec | R1 exec | Inference |
|---|---|---|---|---|---|---|
| **Node** | | Match Rate of Business Entity Formation | – | 70% | 60% | LLMs and Humans Disagree on Formed Entities |
| **Intra** | **Coverage** | 1. Gather client info | 0.74 ± 0.28 | 0.69 ± 0.21 | 0.44 ± 0.17 | Key step for humans with high coverage of all tasks |
| | | 2. Decide Recommendation | 0.63 ± 0.31 | 0.84 ± 0.17 | 0.31 ± 0.17 | Not so key step for humans with many steps skipped compared to O1. R1 differs. |
| | | 3. Draft Memo | 0.45 ± 0.33 | 0.37 ± 0.32 | 0.53 ± 0.41 | Low human and LLM coverage suggests some subtasks are unnecessary. |
| | | 4. Draft Agreement | 0.40 ± 0.32 | 0.96 ± 0.06 | 0.68 ± 0.24 | Humans cover fewer nodes; task plan may be over-specified. |
| | | 5. Tax Treat [Optional] | 0.39 ± 0.44 | 0.85 ± 0.05 | 0.19 ± 0.17 | O1 over-covers tax; humans skip due to contextual simplicity, as does R1 |
| **Inter** | | Longest Common Sequence | 1→2 (80%) 1→2→1 (30%) | 1→2→3→4→5 (100%) | 1→2 (80%) 1→2→3→4 (40%) 1→2→3→4→5 (20%) | LLM strictly follows the plan, while humans deviate based on the scenario, with no consistent pattern across projects or law RAs. |

Table 5: **Analysis across different granularities** shows that human workflows show adaptive, context-sensitive behavior with iterative revisions and nonlinear transitions, while LLMs exhibit rigid, sequential execution with uniform but less context-aware task coverage.

### A.3.1 Granular Analysis of Human vs LLM Execution

In Table 5, we observe that at the node level, the LLMs achieved 60-70% match rate with humans in entity selection, after the execution of the client elicitation stage. This could have implications on the final agreement drafted. At the intra-subtask level, the coverage metric indicates that humans prioritized early-stage actions such as client intake (0.74±0.28), often skipping lower-yield steps. LLMs, meanwhile, allocated effort more uniformly across subtasks, with high emphasis on drafting the operating agreement (0.96±0.06) and tax considerations (0.85±0.05). However, since most scenarios in our roleplay simulations involved minimal tax complexity, which is an artifact of our simulation, the lower human engagement with tax tasks (0.39±0.44) may reflect greater context sensitivity. Humans also showed more iterative execution, with higher Levenshtein distances between plan and execution (52.9±24.8), suggesting more frequent reordering or revision. At the inter-subtask level, human workflows commonly included return cycles and multiple edits, whereas LLMs followed strictly sequential paths. Overall, we observe that reasoning models like Deepseek-R1 aligns more with human intra-level behavior (lower coverage on subtask 2 and 5).

To provide a more holistic view of these execution patterns across individuals and scenarios, we construct Finite State Machines (FSMs) based on observed reasoning paths (Figure 15). FSM visualizations capture the underlying legal workflow language, showing typical state transitions, common loops, and pathway flexibility across scenarios. By abstracting from individual differences, FSMs provide a holistic view of reasoning dynamics and reveal both normative paths and outlier transitions. This broader modeling enables lightweight monitors that can flag statistically atypical steps without enforcing a single correct sequence, encouraging reflection while respecting the interpretive flexibility inherent in legal work.

### A.3.2 Workflow Monitor

The workflow monitor is a data-driven model trained to detect deviations from human-like reasoning in either other human or LLM-generated workflows. We trained a Llama-3.2-1B-Instruct model (Grattafiori et al., 2024) on eight human-generated workflow scenarios to act as a step-level classifier. Given prior context, the model predicts the most probable next step and uses perplexity to estimate how typical a step is, with lower perplexity indicating closer alignment with human behavior. We hold out 2 human workflows as a test set along with 10 LLM generated workflows.

| Step Transition | Z-score | Flag (Deviation) | Interpretation |
|---|---|---|---|
| 2.3.5 Write memo to client → 3.1 Write memo | 0.11 | On track (0–0.5) | Aligned with expected workflow. |
| 5.5.1 Advise on best entity form → 5.5.2 Refer to tax professional | 0.62 | Needs review (0.5–2) | Additional guidance may be needed. |
| 4.1.2.2 Identify various clauses → 5.1 Ask default tax questions | 29.31 | Off track (>2) | Significantly deviates from human workflow; verify step. |

Table 6: **Z-scores** for each workflow step indicate alignment with human behavior. Steps are flagged as "on track," "needs review," or "off track," highlighting whether they align with typical human reasoning, require further examination, or represent unusual transitions.

| Flag (Deviation Range) | Percentage (%) |
|---|---|
| On track (0–0.5) | 67.82 |
| Needs review (0.5–2) | 14.78 |
| Off track (>2) | 17.39 |

| Tool | Avg Use | Projects (%) |
|---|---|---|
| Note Taking | 54.3 | 100 |
| Document Writing | 8.5 | 100 |
| Download Files | 1.6 | 50 |
| Upload Files | 1.0 | 10 |
| Browser Search | 0.0 | 0 |
| Vector Search | 0.0 | 0 |

(a) **Z-score distributions** show that nearly 70% of LLM-generated steps align with human-like sequence planning.

(b) **Tool usage patterns** on the LawFlow data collection tool: Vector search wasn't used, given the limited, familiar document set. Participants primarily used their familiar tools, such as Westlaw, etc, rather than integrated tools. "Avg use" refers to the average number of times a tool was used per project among the projects that utilized those tools.

Next, we define a **deviance score** as the z-score of a conditional step-wise perplexity (perplexity of only the new generated next-step) relative to the human distribution:

$$\text{Deviance Score} = \frac{\text{step perplexity} - \mu_{\text{human perplexity}}}{\sigma_{\text{human perplexity}}}$$

where $\mu_{\text{human perplexity}}$ and $\sigma_{\text{human perplexity}}$ represent the mean and standard deviation of step-wise human perplexities across the human sequences upon which the model is trained. This represents the deviations relative to the distribution of human behavior. Steps with deviance scores *beyond two standard deviations are flagged as unlikely*, while those near zero suggest strong alignment. Moderate deviations may signal ambiguity or areas that require human input. Rather than enforcing correctness, this monitor provides soft signals to prompt reflection and revision, helping AI systems stay on track while supporting flexible, context-sensitive legal reasoning.

### A.3.3 Examples of Workflow Outputs

For each scenario, in addition to a human drafting the appropriate agreement, the LLM is also used to generate an agreement for the scenario in order to enable comparison. The prompt used to assist the LLM in generating the agreement can be found in Figure 20.

LLM-generated and human-generated agreements for these two cases are described:

(1) A case where the *LLM agreed* with the human on the same business entity recommendation and *selected the same template*. Scenario considered : Simulation 19 - Refurbishing Hockey Equipment (Appendix A.2). Snippets of this human-generated and LLM-generated agreement are shown in Figure 21 and Figure 22.

(2) A case where the *LLM agreed* with the human on the same business entity recommendation but *selected a different template* than the human. Scenario considered : Simulation 15 - Lake Bed and Breakfast. The description of the simulation can be found in Appendix A.2. Snippets of this human-generated and LLM-generated agreement are shown in Figure 23 and Figure 24.

### A.3.4 Interpreting Quality: Divergence in Evaluating Agreements

While structural comparisons help illuminate the *how* of legal reasoning, understanding the *why* behind legal choices requires examining how legal workflows are evaluated. Rubric-based LLM assessment frameworks (Figure 17) emphasize formal completeness and surface-level consistency. The evaluation prompts we use for LLM-as-judge evaluation are shown in figure 18 and figure 19. However, these criteria often diverge from the practical, context-sensitive reasoning employed by legal experts. To explore this disconnect, we compared LLM evaluations of human-authored operating agreements with critiques from legal experts and reflections from the student authors themselves.

As shown in Table 8, LLMs frequently flagged omissions that legal professionals defended as intentional, context-appropriate decisions. For instance, critiques about missing operational specifics overlooked established norms; both the expert and the writer emphasized that such details are best handled in ancillary documents rather than in the operating agreement itself. Similarly, concerns about unanimity requirements failed to take into account the strategic value of consensus in small, closely held businesses, where unanimity can encourage communication and mitigate power asymmetries.. These examples reflect a broader pattern: LLMs often apply generalized standards without recognizing domain-specific practices, client intentions, or the trade-offs that shape real-world legal drafting.

That said, LLM evaluations are not without merit. They surface valuable concerns that human drafters may overlook. In one example, the LLM correctly identified a lack of valuation detail in the right-of-first-refusal clause, an omission the legal expert agreed could generate future disputes. It also flagged the absence of explicit dispute resolution provisions, prompting both expert and student reconsideration. These instances demonstrate how LLMs, with their exhaustive and systematic lens, can serve as a valuable second-pass reviewer, catching latent issues even when their evaluative logic is imperfect. Expert assessments also revealed that LLM-generated agreements are not always inferior. Out of the two agreements that the expert evaluated, they judged one of the LLM-drafted agreement to be *preferable* to the student version. The LLM's draft was praised for its greater clarity, completeness, and improved treatment of stakeholder roles, particularly in how it incorporated provisions for children, which the student agreements neglected. While some of the LLM's choices (e.g., voting thresholds) were seen as potentially confusing, the agreement overall was deemed more thoughtfully constructed. This comparison shows that LLMs are capable not only of critique but also of producing high-quality legal drafts, sometimes outperforming novice human writers.

Ultimately, these comparisons highlight that LLMs currently adopt a formalist perspective, privileging structure, coverage, and textual clarity over the more nuanced, judgment-driven reasoning that human experts rely on. Supporting legal professionals in high-stakes contexts will require AI systems to incorporate better human-centered heuristics, striking a balance between rigor and adaptability, and structure and pragmatism. Bridging this gap requires not only more capable models but also training data that captures how real practitioners navigate ambiguity, trade-offs, and evolving client needs -precisely the kind of insight `LawFlow` is designed to provide.

### A.4 From Insights to Actions: Design Principles for Collaborative Legal Assistants

Building directly on the empirical findings in Section 4, we outline a next-generation legal assistance framework that treats AI not as a replacement for lawyers or judges, but as a collaborative partner whose rigor complements human nuance. A key insight from `LawFlow` is that human and AI reasoning and planning often differ in structure, adaptability, and rigor. These differences can be harnessed to improve outcomes rather than being viewed solely as shortcomings. Each design pillar below maps to a concrete observation from our experiments and sketches how legal practitioners and AI systems can collaborate more effectively.

Currently, workflow quality is primarily judged by evaluating the final output, which is the agreement. While this focus on final document quality reflects real-world expectations, it offers a limited view of the broader reasoning and planning process. To capture workflow

| Question | LLM Eval Trade-Offs | LLM (Eval) Rationale | Legal Expert (Eval) | Student (Writer) Defense |
|---|---|---|---|---|
| Question 1: Does the operating agreement **reflect all customer requirements**? | **Tension between formalism and practical norms**: LLM critiques the omission of specific operational details, reflecting a formalistic lens. However, both the expert and writer emphasize that excluding such details is not a flaw but a norm aligned with best practice. This highlights a trade-off between legal completeness and the flexibility favored in actual business structuring. | Score: 3/5. Reasoning: The operating agreement encompasses broad operational and membership elements, but it doesn't explicitly address all specific operational needs ... | I don't agree with the LLM feedback which faults the operating agreement for just providing a governance framework without addressing specific operational details. But those kinds of details would not generally be included in an operating agreement. | No business consists of one document. There will be leases, use agreements etc. I do not see this as a flaw in the operating agreement. The more specific I make the operating agreement in terms of operations, the less flexibility there is and the more potential for conflict concerning the operating agreement. Other documents can be more specific, but I think generality is good here. |
| Question 2: Are there **clauses that might create unnecessary disputes** between stakeholders? | **Correctly flags potential risks, but underplays context**: LLM appropriately identifies concerns with missing valuation mechanisms and unanimity clauses, especially in dispute-prone situations. Yet, it may overemphasize these risks without accounting for context - the writer and expert view unanimity as a deliberate design choice suitable for a two-member LLC. This suggests a nuanced trade-off between structural safeguards and scalability. | Score: 3/5. Reasoning: Several sections might induce disputes, especially around transfer of interests, capital contributions, and the unanimity required for many decisions. The right of first refusal in Section 4.04(a) lacks detail on how the membership interest is valued in conflicts, possibly leading to contention. Potential deadlocks due to unanimity requirements without explicit dispute resolution mechanisms for internal conflicts might also lead to issues | The LLM expresses concern about the unanimity requirements.. But with a 2 member LLC at the start, it's much less a problem.. The criticism of the valuation in the right of first refusal is absolutely accurate, a real problem with the agreement. The dispute mechanism resolution suggestion is worth considering. With 2 members who have equal voting power, deadlock is unfortunately a built-in problem... | I agree that requiring unanimity in decision making can become a problem if the company grows. But it can be an important mechanism for a small venture like this one—you can't have only two or three members who all disagree. It's a potential for conflict, but it's also an incentive for communication and compromise because you have no option to disregard your fellow members' opinions. "There are no solutions, only trade offs." |
| Question 3: Are all **key terms clearly defined** in the document? | **Mixed accuracy in issue spotting**: LLM correctly identifies gaps in definition for some key terms (e.g., "EBITDA multiplier"), though it misjudges others like "capital contribution," which is defined. This reflects a trade-off in automated review : broad coverage and formal spotting versus precision and context sensitivity. | Score: 3/5. Reasoning: The document lacks clarity on concepts like "Capital Contribution," "Pending Membership Interest," and "EBITDA multiplier." | "Capital Contribution" is defined, but EBITDA Multiplier and Pending Membership Interest are not... | "capital contributions" is defined in Section 2... "EBITDA multiplier" and "pending membership interest" might be self-evident... |

Table 8: We compare **evaluations of student-drafted agreements** across five dimensions. The *LLM (Eval) Rationale* column summarizes the LLM's rubric-based evaluation of the agreement. The *Legal Expert (Eval)* column reflects feedback on the LLM evaluation from a law professor, who is an academic with expertise in legal drafting pedagogy but limited real-world experience. The *Law Student (Writer) Defense* column shows the student's rationale for specific drafting choices. The red colored text indicates disagreement with the LLM while the green colored text indicates agreement. The *LLM Eval Trade-Offs* column captures high-level patterns in how the LLM evaluates legal agreements. LLM evaluations tend to emphasize formal completeness, while human reviewers emphasize practical judgment and contextual appropriateness. This illustrates differing evaluation criteria between LLMs and domain-aware humans.

quality more holistically, we propose supplementing this output-based evaluation with process and behavior-oriented metrics that better aid human goals of clarity, efficiency, creativity, and completeness.

Specifically, we propose three metrics:

1. **Decision consistency**, which measures how reliably participants revisit or stick with their earlier choices helping promote both completeness and clarity across the workflow.
2. **Workflow completeness**, which evaluates whether key subtasks are addressed in a logical and thorough sequence, allowing for greater task completeness and process rigor.

3. **Evaluation of point performance**, which focuses on high-impact decision moments that disproportionately affect downstream outcomes. Careful assessment and assistance at these points improve not only task completeness and clarity but also aid creative problem-solving, as participants are encouraged to generate well-justified solutions rather than relying on rote templates. For instance, selecting the appropriate business entity structure (e.g., an LLC vs. a C-corp) is a pivotal choice that affects liability, taxation, and document organization. An accurate and well-supported decision at this junction enhances clarity throughout the agreement, improves drafting efficiency by reducing revisions, and enables creative structuring options that better align with client goals. By emphasizing point performance alongside overall process quality, we can better assess where strong legal judgment, rather than just procedural diligence, drives superior results.

We outline design principles (illustrated in Figure 13) aligned to these metrics, each aimed at improving legal workflows through human-AI collaboration.

**1. Collaborative Planning – Human Depth + AI Breadth**    We observe that humans build deep, modular plans, while LLMs generate exhaustive but rigid checklists. Rather than judging one style as superior, we can merge their strengths.

- *Human skeleton.* The lawyer sketches 3–6 top-level goals, mirroring the deep, modular structures observed in human plans.
- *AI breadth pass.* AI expands each goal into a collapsible checklist, surfacing edge cases the lawyer may overlook, leveraging the AI's exhaustive but flat planning style.
- *Reconciliation view.* A side-by-side diff lets the user prune or promote AI branches, yielding a jointly developed hybrid plan.

This intuition of mutual support aligns with lessons from judicial decision-making research (Spamann & Klöhn, 2016; 2024), which shows that while human judges often deviate from strict legal precedent due to extralegal factors, such as sympathy, LLMs exhibit a more rigid, formalist approach similar to that of law students. Whether formalism is an asset or a liability depends on the task at hand. In drafting and compliance, a systematic approach can help reduce overlooked details, but in complex, high-stakes judgments, the human judge's nuanced weighing of facts and equities may be indispensable. This hybrid design could enhance both **workflow completeness** and **evaluation of point performance** by encouraging lawyers to combine structured human reasoning with AI-surfaced contingencies. It directly supports the human goal of task completeness, helping lawyers structure more thorough and resilient workflows without sacrificing focus.

**2. Adaptive Execution Engine & Reflective Monitoring**    During execution, law students routinely *loop back* and reorder tasks, whereas LLMs predominantly execute through a plan once, in strict sequence. To support human adaptability without sacrificing coherence, we propose:

1. *Live plan mutation.* When new facts emerge, users can drag and rearrange tasks; the LLM automatically rethreads downstream dependencies, maintaining coherence while allowing dynamic adjustment.
2. *Workflow-diversity monitor.* A lightweight model flags statistically unusual moves relative to prior human traces, prompting reflective checks without enforcing strict conformity.

These tools could streamline plan adjustments and spotlight deviations, promoting greater efficiency and clarity. By supporting **decision consistency** and **workflow completeness**, they help maintain coherence while still allowing creative flexibility. Reflection could also be deepened through lightweight nudges from LLMs, such as surfacing stale assumptions or offering comparative junction-specific peer views.

**3. Meta-Decision Point Assistance**    In our work, betweenness centrality and time-spent analyses identify note-taking, client-elicitation questions, entity selection, and reviewing operating agreements as high-leverage nodes that impact the workflow of the drafting

process. Thus, we propose design suggestions that modify the LLM's role at these meta-decision points.

1. *Front-loaded prompts.* During client intake, the assistant auto-suggests clarifying questions and surfaces similar fact patterns, ensuring critical details are captured before substantive work begins.
2. *Review-only drafting.* In later stages, the LLM defaults to "redline & comment" mode, offering critiques and consistency checks unless the user explicitly requests auto-rewrites.

Thus, AI effort is focused exactly where it can significantly aid the lawyer, and the system could enhance human creativity (by encouraging thoughtful exploration of alternatives) and completeness (by ensuring pivotal issues are surfaced early and handled carefully). This strategy directly strengthens **point performance**, ensuring workflows are anchored by strategically sound choices.

Beyond the immediate design suggestions, the `LawFlow` dataset also can be synthetically extended into multiple "near-miss" variants by reordering steps or omitting low-value actions. These chain-of-thought (COT) style expansions would multiply the dataset while preserving a realistic, student-level voice, enabling richer modeling of developmental reasoning. Such extensions open the door to broader outcomes, including peer-mentoring systems, scaffolded checklists, and alternative-path exemplars. Together, these developments could create a living library of legal workflows, helping users not only complete tasks more effectively but also continuously strengthen their reasoning and drafting strategies in ways grounded in authentic legal practice.

In this broader collaborative framework, AI-supported tools built around `LawFlow`-style metrics do more than promote formal completeness and decision consistency, they accelerate drafting workflows, improve document clarity and robustness, and spark creativity by surfacing overlooked opportunities and trade-offs. Rather than replacing human expertise, such systems scaffold better professional judgment, helping legal practitioners produce work that is both technically sound and strategically optimized.

```
Step 1
<step_analysis>
.
.
</step_analysis>

<knowledge_assessment>
   .
   .
</knowledge_assessment>

<lawyer_action>
The lawyer blocks off time to speak with two experienced colleagues in the firm's
business-formation group. In a confidential setting, the lawyer outlines the broad
outlines of the matter - three individuals pooling resources for a healthcare practice,
with one non-physician demanding equal equity but bringing primarily sweat equity. The
lawyer highlights concerns about diverging interests and invites feedback on:

- Whether representing all three individuals raises insurmountable conflicts of interest
- or if representing the entity might be more suitable.
- How other attorneys have handled situations where a key administrative partner wants a
significant stake despite limited capital.
- Applicable state regulations or ethical considerations related to professional service
entities mixing physician and non-physician ownership.

One colleague suggests reviewing the state's professional corporation and LLC statutes to
 see if non-physician ownership holds any special limitations. Another colleague
emphasizes the sensitivity of the doctors' non-compete constraints and advises clarifying
 how patient follow-up will occur. Both recommend the lawyer consider sending each client
 for independent counsel or at least addressing the conflict-of-interest disclosures in
writing.

The lawyer takes detailed notes, planning to incorporate the colleagues' suggestions into
 the next steps, including verifying local healthcare licensing rules on non-doctor
ownership, reevaluating how to proceed with joint representation (if feasible at all),
and preparing to speak with the clients about potential alternative structures. The
lawyer concludes by thanking the colleagues and scheduling a follow-up to share any
further updates once new information is gathered.
</lawyer_action>

<next_suggested_step>
2.1 Think about recommendations
</next_suggested_step>

Step 2
...
<lawyer_action>
The lawyer sits down at their desk and begins organizing the considerations drawn from
both legal research and previous colleague consultations. They jot down a list of
possible recommendations that address ownership structure, management authority, and
compliance with state professional practice requirements.

First, the lawyer reviews the state laws that govern professional service entities. They
note that some states allow a non-physician to hold a partial stake in a medical practice
, but with restrictions on certain decision-making powers related to clinical care. The
lawyer sketches out possible corporate forms (e.g., forming a professional limited
liability company versus a professional corporation) and highlights how each structure
might impact management rights and liability coverage. They also reflect on conflict-of-
interest rules and the duty to inform the potential co-owners about the lawyer's role.

Next, the lawyer drafts a series of bullet points for each recommended approach to
ownership and governance:

.
.

Finally, the lawyer places a star next to the key question of how to handle the potential
 conflict of interest in representing all three. The lawyer contemplates whether to send
short disclaimers or have an upfront conversation with each client. The next step, the
lawyer decides, is to weigh the pros and cons of each ownership approach and prepare a
concise outline to discuss with the clients.
</lawyer_action>

<next_suggested_step>
2.1.1 Consider counter-arguments/pros and cons
</next_suggested_step>
```

Figure 11: LLM Execution Outputs showing 2 consecutive steps generated

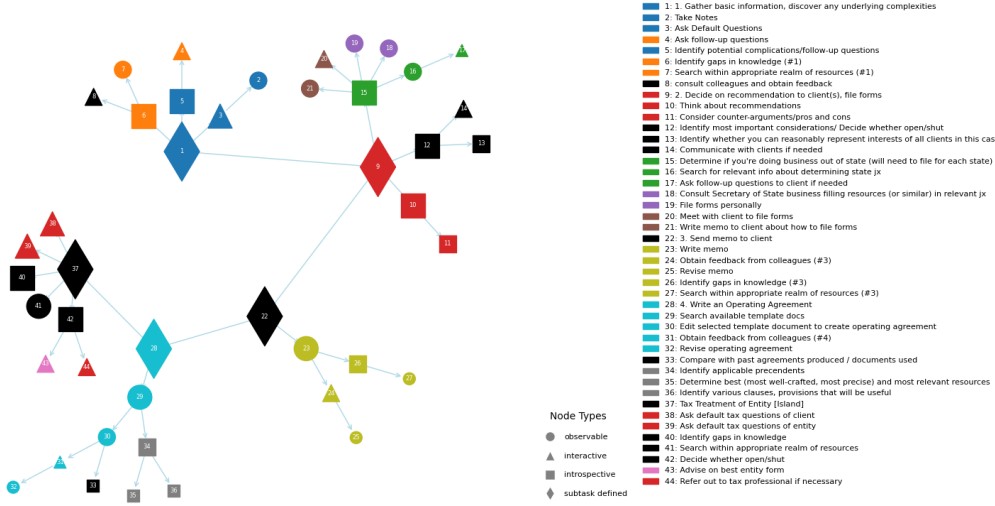

(a) Human Task Plan, color mapped to LLM Task Plan

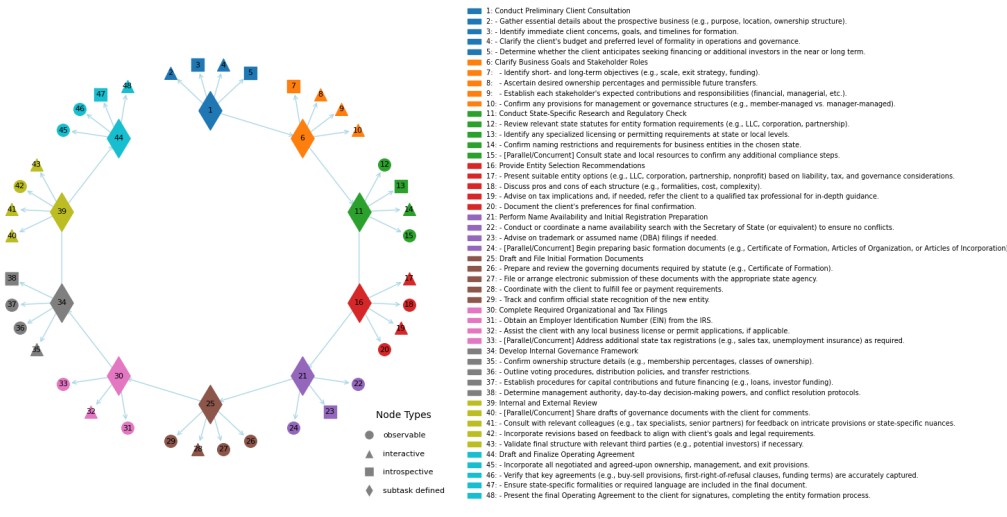

(b) LLM Task Plan

Figure 12: This figure shows the mapping of all subtasks from the LLM Task Plan onto the Human Task Plan. Black nodes in the Human Task Plan represent steps that exist in the Human Task Plan but have no corresponding match in the LLM Task Plan, indicating that LLM-generated plans omit these steps. Additionally, multiple LLM subtask nodes often map under a single human subtask node, suggesting that human plans are more modular and hierarchically structured than LLM plans.

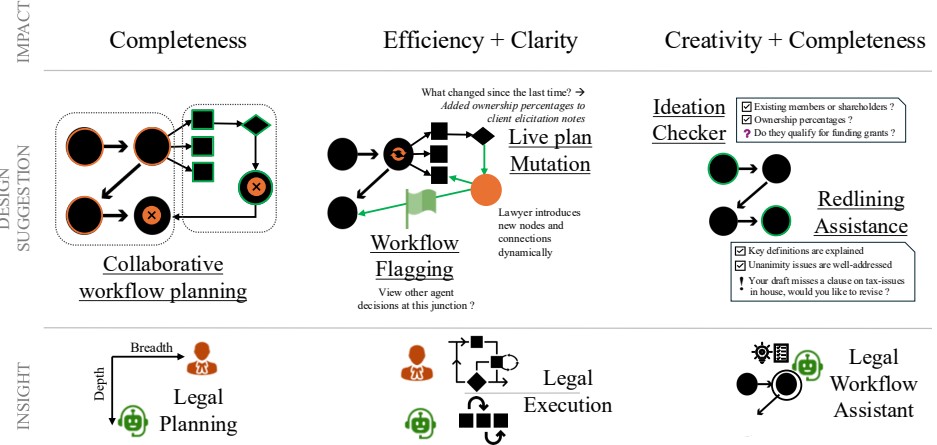

Figure 13: **Design suggestions** based on insights learned from LawFlow include collaborative workflow planning, workflow flagging and AI assistance with specialized roles which can aid human goals of clarity, efficiency, creativity and completeness.

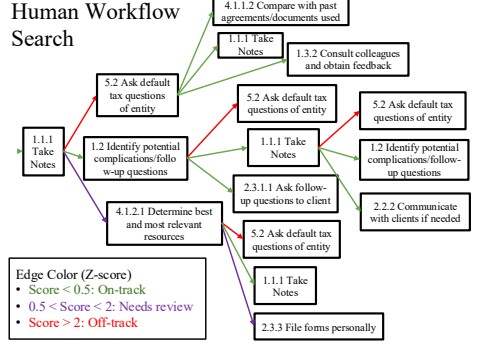

(a) **Human workflows** tends to favor commonly observed subtask sequences, such as note-taking followed by clarification and follow-up steps.

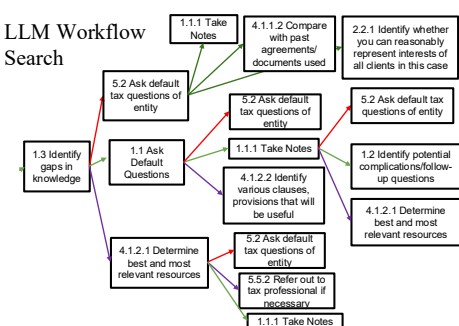

(b) **LLM workflows** show jumping across task categories or introducing less context-sensitive steps, such as prematurely selecting resources or reverting to earlier-stage tasks.

Figure 14: **Step-level beam search visualizations** of Human and LLM workflows, colored by atypicality score (z-score) as predicted by a our workflow monitor : Transitions are scored by how well they align with human reasoning norms. Human workflows follow stable, commonly observed reasoning patterns, while LLM workflows exhibit greater variability and introduce less context-sensitive transitions.

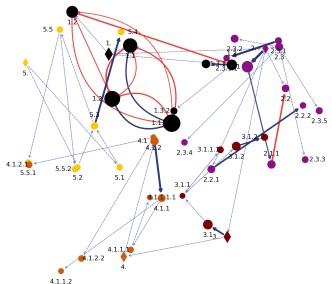 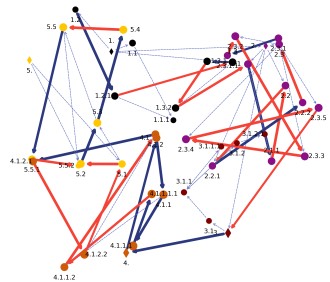 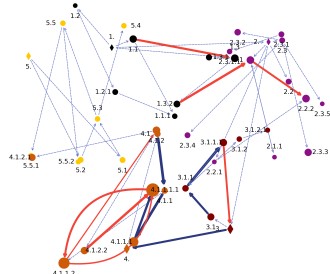

(a) **Human FSM**: show higher variability in path selection and frequent revisiting of prior states. Larger node sizes indicate multiple re-engagements with key tasks, and sparse lateral connectivity suggests a tendency to follow one branch deeply rather than exhaustively traversing all parallel paths.

(b) **GPT O1 FSM**: reveals a highly exhaustive traversal style, covering nearly all nodes, including sibling tasks on the same hierarchical level. Because the task plan omits horizontal (same-level) transitions, O1's breadth-first traversal across sibling nodes results in many red edges.

(c) **Deepseek R1 FSM**: exhibits a more restrained traversal than O1, engaging fewer sibling nodes and showing a transition pattern more consistent with depth-oriented exploration. The FSM displays wider coverage than human paths, but fewer red edges than O1, suggesting a hybrid strategy that partially aligns with human planning structure.

Figure 15: **FSM visualizations** highlight differences in aggregate execution dynamics between humans and LLMs. *Node size* indicates frequency of revisits; *edge thickness* reflects transition probability. *Blue edges* denote planned transitions in task plan; *Red edges* represent executed transitions not in the task plan. *Solid edges* were executed; *Dashed edges* were planned but not followed. We only display edges which have a threshold $> 0.5$. Human workflows show selective, depth-oriented exploration; O1 follows an exhaustive, breadth-first pattern with many lateral transitions; R1 takes a more focused approach, partially resembling human strategies.

(a) Comparing the time spent on different cognitive tasks per RA

(b) Comparing the time spent on different sub-tasks in the task plan per RA

Figure 16: **Comparison of activity timelines** grouped by sub-tasks and cognitive task types across all Law RAs

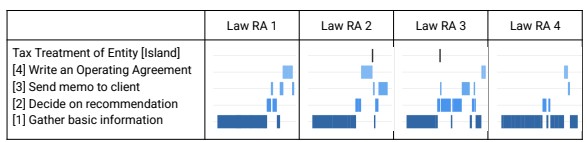

Figure 17: Rubric for Evaluating Operating Agreements

```
## Answer these questions based on the following scenario description and its operating
agreement. Provide a score out of 5 for each question, along with reasoning for the score
.

### Formal

- Are all parts of the operating agreement internally consistent? (Score: X/5, Reasoning:
 ...)

- Are all key terms clearly defined in the document? (Score: X/5, Reasoning: ...)

- Is the language clear and precise? (Score: X/5, Reasoning: ...)

### Substantive
- Does the operating agreement reflect all customer requirements? (Score: X/5, Reasoning:
 ...)

- Is the operating agreement practical and enforceable? (Score: X/5, Reasoning: ...)

- Are there clauses that might create unnecessary disputes between stakeholders? (Score:
X/5, Reasoning: ...)

- Does the operating agreement capture the intended structure and goals of the business?
(Score: X/5, Reasoning: ...)

- Will this operating agreement make it difficult for stakeholders to exit or dissolve
the entity if needed? (Score: X/5, Reasoning: ...)

### Scenario Description
Title: {TITLE}
Description: {DESCRIPTION}

### Operating Agreement}
{OA}

### Answer
```

Figure 18: LLM-as-Judge general questions evaluation prompt

```
## Answer these questions based on the following scenario description and its operating
agreement. Provide a score out of 5 for each question, along with reasoning for the score
.

### Formal
- Does it follow the minimal standards of an operating agreement checklist? (Score: X/5,
Reasoning: ...)

#### Operating Agreement Checklist
{CHECKLIST}

### Scenario Description
Title: {TITLE}
Description: {DESCRIPTION}

### Operating Agreement
{OA}

### Answer
```

Figure 19: LLM-as-Judge LLC Checklist question evaluation prompt

```
You are a lawyer. Your role is to analyze the provided business scenario, determine the
most suitable business entity to form for your clients, and select the most appropriate
template from a given list to draft a complete, legally enforceable governing document.

### Instructions
Analyze the given business scenario, determine the most suitable business entity type,
choose an appropriate template from the provided list, and draft a complete, legally
binding document.
### Scenario Title
{TITLE}
### Scenario Description
{DESCRIPTION}
### Available Templates
Template 1
[...]
Template 2
[...]
### Business Entity Recommendation, Template Selection and Writing an Agreement
```

Figure 20: LLM Agreement Generation Prompt

```
OPERATING AGREEMENT OF MADSTICKS LLC.

This Operating Agreement (this ``Agreement'') is entered into by and between Brendan
Madson [...]

Name.  The name of the limited liability company is Madsticks LLC [...]

Assumed Name.  The Company may do business under an assumed name from time to time, as
provided under Chapter 322C of the Minnesota Statutes [...]

Purpose.  The purpose and character of the business of the Company shall be to undertake
and carry on any lawful business, purpose [...]

Operating Agreement.  This Agreement constitutes the ``Operating Agreement'' of the
Company as defined in Section 322C.0102, [...]

Principal Place of Business.  The principal business office of the Company will be
located at 123 Hockey Lane, St. Paul, [...]

Registered Office and Registered Agent.  The location of the registered office and the
name of the registered agent (if any) of the Company [...]

No Personal Liability.  The liability of the Sole Member shall be limited to the fullest
extent permitted by law [...]

Capital Contributions; Membership Interest in the Company.  The Sole Member's capital
contribution is set forth on the attached Schedule A [...]

Additional Contributions.  The Sole Member is not required to make any additional capital
 contributions to the Company [...]
```

Figure 21: Operating Agreement drafted by Human (*LLM agreed with the human* on the same business entity recommendation and selected the same template.)

```
# OPERATING AGREEMENT
OF
L's Refurbished Hockey Equipment LLC

This Operating Agreement (this ``Agreement'') is entered into by and between L [Last Name
] (the ``Sole Member'') and L's Refurbished Hockey Equipment LLC, a Minnesota limited
liability company (the ``Company''), as of the Effective Date (as defined below)[...]

1. **Name.** The name of the limited liability company is L's Refurbished Hockey
Equipment LLC.

2. **Assumed Name.** The Company may do business under an assumed name from time to time,
 as provided under Chapter 322C of the Minnesota Statutes[...]

3. **Purpose.** The purpose and character of the business of the Company shall be to
undertake and carry on the business of refurbishing used hockey equipment[...]

4. **Operating Agreement.** This Agreement constitutes the ``Operating Agreement'' of the
 Company as defined in Section 322C.0102,[...]

5. **Principal Place of Business.** The principal business office of the Company will be
located at [Your Home Address], or such location as may hereafter be determined by the
Sole Member.

6. **Registered Office and Registered Agent.** The location of the registered office and
the name of the registered agent (if any) of the Company[...]

7. **No Personal Liability.** The liability of the Sole Member shall be limited to the
fullest extent permitted by law[...]

8. **Capital Contributions; Membership Interest in the Company.** The Sole Member's
capital contribution is set forth on the attached Schedule A[...]

9. **Additional Contributions.** The Sole Member is not required to make any additional
capital contributions to the Company.[...]
```

Figure 22: Operating Agreement drafted by LLM (*LLM agreed with the human* on the same business entity recommendation and selected the same template.)

```
OPERATING AGREEMENT
[COMPANY]

This Operating Agreement (this ``Agreement'') of Aunt Doris's B&B, L.L.C., a
Minnesota limited liability company (the ``Company''), is entered into as of December 30,
 2024  (the ``Effective Date'') by and among the Company, Robin, and Daniel (each a ``
Member'' and collectively the ``Founding Members''), and any other person or entity that,
 after the date above, becomes a Member of the Company in accordance with the terms of
this Agreement.

ARTICLE 1
Organizational Matters

Section 1.

Name. The name of the Company is Aunt Doris's B&B, L.L.C.

Section 2.   Assumed Name; Business Name. The Company may do business under an
assumed  name  from  time  to  time,  as  provided  by  the  Minnesota  Revised  Uniform
 Limited
Liability  Company  Act,  Minn[...]

Section 3.

Principal  Office.  The  principal  office  of  the  Company  is  located  at
[ADDRESS],[...]

Section 4.

Registered  Office;  Registered  Agent.  The  registered  office  and  agent for
service  of  process  on  the  Company,  in  the  State  of  Minnesota,[...]

Section 5.

Purpose; Powers.

(a)  The  purpose  of  the  Company  is to engage in any lawful business purpose or
activity in accordance with the MN RULLCA[...]
```

Figure 23: Operating Agreement drafted by Human (LLM agreed with the human on the same business entity recommendation but *selected a different template* than the human)

```
OPERATING AGREEMENT
OF
Lake Bed and Breakfast LLC

TABLE OF CONTENTS

Article 1 Definitions    1
Article 2 Formation      4
Article 3 Capital Contributions        5
Article 4 Allocations of Profits and Losses; Distributions     6
Article 5 Management     7
Article 6 Books and Records; Tax Matters        11
[...]

THE LIMITED LIABILITY COMPANY INTERESTS (OR ``UNITS'') OF THE COMPANY DESCRIBED IN AND
GOVERNED BY THIS AGREEMENT HAVE NOT BEEN REGISTERED UNDER THE SECURITIES ACT OF 1933, AS
AMENDED, OR UNDER ANY APPLICABLE STATE SECURITIES LAWS. THE UNITS ARE RESTRICTED
SECURITIES WITHIN THE MEANING OF RULE 144 PROMULGATED UNDER THE SECURITIES ACT OF 1933,
AS AMENDED[...]

OPERATING AGREEMENT
OF
Lake Bed and Breakfast LLC

This OPERATING AGREEMENT (``Agreement'') is made this ______ day of _______, 202__, by
and between M and A and their three children, collectively, the ``Members'' and each,
individually, a ``Member''.

Recitals
The undersigned constitute all of the current Members of the Company.
Each of the undersigned desires to enter into this Agreement, [...]
Agreement
In consideration of the foregoing and the mutual promises and agreements set forth below,
 the Members agree as follows:

### Article 1. Definitions

The terms defined in this Article 1 [...]
``Act'' means the Minnesota Revised Uniform Limited Liability Company Act [...]

``Additional Member'' means a Person who is admitted as a Member and issued a new Company
 Interest.

``Affiliate'' means, with respect to any Person, (i) any Person that directly or
indirectly through one or more intermediaries controls or is controlled by or is under
common control with the specified Person, [...]
```

Figure 24: Operating Agreement drafted by LLM (LLM agreed with the human on the same business entity recommendation but selected a *different template* than the human)

