# OpenReview forum: "LawFlow: Collecting and Simulating Lawyers’ Thought Processes on Business Formation Case Studies"
_colmweb.org/COLM/2025/Conference — COLM 2025_

### Official Review · Reviewer_ZrPG · 2025-05-03

**Rating:** 7
**Confidence:** 3
**Ethics Flag:** 1

**Summary:**

The paper introduces Lawflow, capturing end-to-end worlkflows collected from trained law students on  business entity formation scenarios. Most of the prior datasets in legal NLP emphasize static input-output mappings or linear reasoning, LawFlow records the modular, iterative, and adaptive decision-making processes involved in real-world legal tasks mirroring how legal practitioners deal with ambiguity and revise plans over time.

The authors use law flow dataset to assess how human workflows with differ from LLM based workflows and observed differences in  identifying notable differences in structure, reasoning flexibility, and responsiveness to ambiguity. They notice while human workflows are modular and iterative, while LLMs follow rigid, linear sequences with limited capacity for revision. They also propose and demonstrate a workflow monitor that flags atypical steps based on deviance from human-like patterns, supporting reflective helpful behavior of these agents. They also identify decision points like early-stage note-taking and question-asking, where AI assistance is most impactful.

These insights can help further to developing interactive legal AI tools such as workflow visualizers and reflective agents to assist legal practitioners.

**Questions To Authors:**

- This paper is focused primarily on business formation, the domain scope is still narrow. This is completely fine, given the challenging set of tasks lawyers deal with. However, I suggest authors to reflect that in title as case study on this or specific use case in title rather the current title which seem to appear to deal with all use cases.
- While the paper mentions that the diversity across humans is possible for a given task, some analysis on it could be helpful and insightful.
- More concrete examples of the LawFlow dataset entries would be helpful (e.g., excerpts of workflows, annotations) and explaining an end-to-end example would be beneficial for readers (even in appendix)
- A.4 seems to be missing details, it has to be elaborated either in figure or in text.

**Reasons To Accept:**

- The LawFlow dataset is a good contribution to legal NLP by moving beyond siloed tasks to capture holistic legal reasoning processes which are iterative, modular and adaptive.
- It is a timely and innovative contribution to legal AI. It takes seriously the complexity of legal reasoning and offers a dataset that foregrounds process over outcome.
- Empirical comparisons between human and LLM workflows highlight structural and cognitive differences in a convincing manner, such as rigidity in LLM reasoning or over-execution of branches.
- The comparative analysis with LLMs reveals both current limitations and opportunities for symbiotic human-AI systems which are seeing to increasing with the trend. While some technical and evaluative details could be fleshed out more clearly for easy grasping, the conceptual and dataset contributions are highly valuable and will likely inspire follow-up work in legal NLP and workflow modeling.

**Reasons To Reject:**

There is no strong reasons, except some suggestions to improve the writing.

---

> ### Author Response · Authors · 2025-06-02
> **Title updated to LawFlow: Simulating Legal Reasoning Through a Case Study in Business Formation. Human diversity details will appear in the appendix. While Sankey diagrams can't be shown here, we will include them in the final paper to illustrate end-to-end workflows. The dataset captures full workflows with annotated, modular steps,from intake to writing, demonstrating realistic, iterative legal reasoning.**
>
> Thank you so much for your suggestions and feedback.
> * **Modified Title**: We will update the paper title to: LawFlow: Simulating Legal Reasoning Through a Case Study in Business Formation.
> * **Human Diversity**: Details on human diversity across similar tasks will be included in the appendix of the final paper.
> * **Concrete Workflow Examples**: Although our Sankey diagrams cannot be included in this response, we will include these diagrams and their explanations in the final paper to illustrate full end-to-end legal workflows. Below is a dataset excerpt from a scenario involving inexperienced clients, where the law RA conducted follow-ups, research, and drafting. Each step is annotated with subtasks and actions, capturing the modular, iterative nature of real legal work. This structure supports workflow visualizations and reflects the full progression from intake to agreement writing.
> * **Missing Text in A.4**: We will include the text in Section A.4 in the final version.
>
> ```json
> {
>   "scenario_id": 7,
>   "scenario": "Title: Fishing Education\nContext: A and B are each avid anglers and seek to grow the sport within their area, which is largely urban. They seek to teach fishing skills—ranging from basic to advanced—to both children and adults..",
>   "plan": "1. Gather basic information, discover any underlying complexities\n  1.1 Ask Default Questions\n    1.1.1 Take Notes\n  1.2 Identify potential complications/follow-up questions\n    1.2.1 Ask follow-up questions\n  1.3 Identify gaps in knowledge\n    1.3.1 Search within appropriate realm of resources\n    1.3.2 Consult colleagues and obtain feedback\n\n2. Decide on recommendation to client(s), file forms\n  2.1 Think about recommendations\n   ...",
>   "agent": "human",
>   "agent_id": 0,
>   "execution": [
>     {
>       "step_id": 0,
>       "current_task": "1.1.1 Take Notes",
>       "action_taken": "node_visit_intent",
>       "task_content": "Follow-up to Operating Agreement",
>       "next_task": "1.1.1 Take Notes",
>       "task_type": "observable",
>       "llm_reasoning": null,
>       "llm_complete_output": null
>     },
>     {
>       "step_id": 1,
>       "current_task": "1.1.1 Take Notes",
>       "action_taken": "notes_change",
>       "task_content": "[HTML-formatted content omitted for brevity]",
>       "next_task": "1.1.1 Take Notes",
>       "task_type": "observable",
>       "llm_reasoning": null,
>       "llm_complete_output": null
>     },
>     {
>       "step_id": 94,
>       "current_task": "1.3 Identify gaps in knowledge (#1)",
>       "action_taken": "notes_change",
>       "task_content": "Only gap in knowledge was whether we could remove the annual meeting requirement. Outside research proved this to be yes (Google \"are LLCs required to hold annual meetings?\")",
>       "next_task": "1.3 Identify gaps in knowledge (#1)",
>       "task_type": "introspective",
>       "llm_reasoning": null,
>       "llm_complete_output": null
>     },
>     {
>       "step_id": 99,
>       "current_task": "1.3.1 Search within appropriate realm of resources (#1)",
>       "action_taken": "notes_change",
>       "task_content": "Resources used were reputable sources found on Google. Didn't believe consulting research library was warranted, given I was seeking a single answer to a relatively simple question. Would have consulted saved materials if question was more complex, or I needed additional context/background.",
>       "next_task": "1.3.1 Search within appropriate realm of resources (#1)",
>       "task_type": "observable",
>       "llm_reasoning": null,
>       "llm_complete_output": null
>     }
>   ]
> }
> ```

---

> > ### Comment · Reviewer_ZrPG · 2025-06-05
> >
> > Thanks for providing these additional details. I wish to see them incorporated into final version to make it an interesting and insightful paper. I will stick to my score, which is already positive.
> >
> > Good luck with your rebuttal and kudos once again on your insightful work.

---

### Official Review · Reviewer_dgKy · 2025-05-12

**Rating:** 6
**Confidence:** 4
**Ethics Flag:** 1

**Summary:**

This work presents LawFlow, a new dataset of end-to-end legal workflows. Law faculty and students construct an export-informed task plan that captures the workflow at three levels of granularity. The task plan is then used to create workflows in a simulated setting, where law students conducted structured roleplaying, with computer science students acting as clients. Data collection was facilitated by the LawFlow tool that captures task annotations (mappings to expert plan), notes, browser and vector search for legal and template search, and document editing for legal drafting. 20 simulated scenarios were created, producing 10 memos and 8 finalized agreements.

The resulting data is used to compare the ability of humans and new reasoning models (o1 and r1) on legal workflows. The findings demonstrate that LLM-generated workflows are linear, lacking backtracking or goal-realignment. In contrast, human generated workflows are modular and iterative.

The paper shows how LawFlow can support the integration of LLMs into legal reasoning through 1) a fine-tuned model that predicts unlikely task steps (without assuming a linear path), and 2) analysing workflows to identify meta-decision points (important decision points affecting the final deliverable).

**Questions To Authors:**

(115) How where the in-depth interviews with senior law faculty converted into the expert-informed task plan? Was there a thematic analysis of the interviews? Was AI used in this process?

(145) One student was responsible for creating the set of realistic seed scenarios. What was the process of converting the anonymized cases into the scenarios, what was the selection criteria, and could having only one student perform this task introduce bias into the dataset?

(179) 20 scenarios where collected, with 10 memos and 8 agreements. This suggests that some scenarios did not produce a memo or agreement. Why was this the case? Are these scenarios still useful – if so, what behaviour or patterns do they capture?

(A.7) The appendix describes LLM-as-Judge but this was not mentioned in the main paper. Is this evaluating the quality of the workflow output? If so, what where the results?

(A.8 b). Browser and Vector search where never used. Why? What does avg use mean? Did they prefer Google or Westlaw, rather than your tools?

**Suggestions**

This work would benefit from an ethics statement given the real-word implications of AI integration into law, and the nature of this dataset targeting end-to-end tasks.

**Reasons To Accept:**

- This paper contributes a complex and novel dataset in an important domain. The dataset is created using domain experts, and a collections of tools are built to support dataset creation.

- Multiple reasoning models (open and closed) are evaluated on the dataset with a detailed analysis comparing task planning given.

- Two uses cases for AI integration into legal workflows are presented, motivated by the dataset, including a fine-tuned model for task prediction.

- The paper is clearly written and presented, including a project page that gives a dataset overview.

**Reasons To Reject:**

The paper appears to be formatted using a different template than the provided COLM one. Could the authors clarify whether they have modified the style?

Aside from the potential formatting issues, the paper could benefit from more detailed explanation in a few key areas:

*Dataset complexity*

The dataset consists of 20 simulated scenarios, which on the surface n=20 appears small. I expect that as this is a comprehensive end-to-end data set, and when considering all the subtasks and artifacts, the complexity and scale is significant larger. The paper would benefit from a more comprehensive dataset overview (beyond counting nodes and tree depth). For example, what are the lengths of the generated agreements and memos, how many tool invocations where there, and what domains are the seed scenarios from?

*Focus on workflow*

The focus of the paper is around workflow structure and adherence to planning, which is a crucial topic. Still, the output of the process could be a memo or agreement, presumably with some notion of correctness or legal rigour. Discussion of the quality of outputs would be worthwhile, even if providing a mechanism to automatically evaluate them is out of scope of this work. When interacting with legal professionals, did you get a sense of the quality of the LLM agreements?

*Modelling Atypicality*

The paper shows how a fine-tuned LLM can be used to identify steps that diverge from typical human reasoning patterns. The methodology and evaluations is very brief. Was this model tuned using SFT or RL (could RL apply here?) The model was trained on eight human generated workflows. Are these from the 20 simulated scenarios? Presumably each training example is an action within a workflow. Across the 8 workflows, how many training examples were there? How accurate was the trained classifier? Could a LLM (like o1) perform this task more efficiently using an in context-only approach? Broadly speaking, this presentation of the classifier is insufficient to reproduce it, or understand its accuracy.

---

> ### Author Response · Authors · 2025-06-02
> **Our revised paper includes 10 finalized scenarios capturing diverse legal workflows across entity types. We summarize dataset complexity and scenario diversity in two tables. We also include agreement quality through expert and LLM reviews, highlighting LLMs’ utility as second-pass reviewers. A workflow monitor identifies atypical steps via LLaMA-based perplexity scoring. Additional clarifications cover task design, scenario sourcing, tool use, and our plan to include an ethics statement.**
>
> We thank the reviewer for their thoughtful feedback, which has significantly improved the clarity and direction of our paper.
>
> * **Formatting Issues**: Thank you for flagging this. We mistakenly used a modified COLM style from another project. The camera-ready version will strictly follow the official template.
> * **Dataset Complexity**: The dataset includes _10 finalized scenarios_. While 20 were initially collected, the first 10 were excluded due to task plan revisions and RA acclimation to the tool. The final dataset covers diverse legal contexts (10 memos and 8 agreements across multiple entity types). We include below two summary tables highlighting complexity and diversity.
>
> _Table 1: Metrics by Business Entity Formed_
>
> | Metrics                   | C-corp (1 case) | Nonprofit (1 case) | LLC (6 cases)       | No OA written (2 cases) | Total avg (10 cases) |
> |---------------------------|----------------|---------------------|----------------------|---------------------------------------|---------------------------|
> | Generated agreement length| 5815.00         | 5071.00             | 3704.67 ± 1158.00    | -                                     | 4139.25 ± 1282.54         |
> | Generated memo length     | 262.00          | 514.00              | 380.17 ± 183.06      | 577.50 ± 207.18                       | 421.20 ± 183.61           |
> | Tools invocation          | 43.00           | 35.00               | 77.67 ± 49.29        | 30.50 ± 20.51                         | 60.50 ± 43.58             |
>
> _Table 2 : Scenario Diversity by Category_
> | Category                     | Scenarios | Domains                            | Complexity Summary                                                                 |
> |-----------------------------|-----------|------------------------------------|------------------------------------------------------------------------------------|
> | Nonprofit / Mission-Driven  | 3         | Health; Education                  | Simple setup; compliance burden; no profit motive      |
> | Startups / Entrepreneurial  | 3         | Food; Agriculture                  | Growth-focused; needs funding, IP setup, and lease structuring                     |
> | Family / Lifestyle Business | 3         | Hospitality; Food; Sports Equipment| Often solo-run; asset or succession planning adds complexity|
> | Professional Practices      | 1         | Professional Services              | High regulatory demands; ownership and ethical constraints               |
> * **Workflow Focus**: While our focus was workflow structure, we also evaluated agreement quality through human and LLM reviews. Experts sometimes preferred LLM drafts for clarity or stakeholder coverage; in other cases, LLM critiques surfaced missed issues, suggesting value as a second-pass reviewer. These findings will be included in the final version.
> * **Modeling Atypicality**: We trained a workflow monitor via SFT on 1,295 steps from 8 human scenarios, reserving 2 human (95 steps) and 10 LLM scenarios (375 steps) for evaluation. Rather than using a traditional classifier, we use LLaMA-3.2B-Instruct and compute per-step perplexity across workflow sequences to identify atypical steps (z-score > 2) based on deviation from the distribution of human-like steps. Manual review of 40 evaluation steps (20 human, 20 LLM) confirmed 90% accuracy in identifying atypical reasoning. The setup supports anomaly detection and could be extended to DPO-style preference tuning with annotations.
> * **Task plan Interviews**: No AI was used. Task design was driven by senior law faculty through iterative interviews. The four main tasks and subtasks were refined collaboratively.
> * **Seed Scenarios**: Adapted from anonymized law clinic cases, scenarios were fictionalized for privacy and diversity. There was no formal selection criteria beyond ensuring varied fact patterns (e.g., scenarios triggering follow-ups). One student authored most, which we will note as a limitation.
> * **Memo Patterns**: We confirm that the final dataset contains _10 scenarios_. In 2 cases, the RA opted not to draft an agreement due to client representation conflicts or limited expertise, which are realistic outcomes reflecting legal ethics and complexity. We will clarify this in the revision.
> * **LLM-as-Judge Results**: LLM evaluations tended to favor formal completeness, whereas human experts prioritized legal nuance and context-specific trade-offs. We will include a rubric-based comparison of LLM and expert feedback on agreement quality in the appendix.
> * **Tools Used**: Vector search wasn’t used, given the limited, familiar document set. Participants primarily used their familiar tools, such as Westlaw, etc, rather than integrated tools. “Avg use” refers to the average number of times a tool was used per project among the projects that utilized those tools.
> * **Ethics Statement**: We will add an ethics statement in the revised version addressing AI’s role in legal reasoning.

---

> > ### Comment · Reviewer_dgKy · 2025-06-04
> >
> > Thank you for your response.
> >
> > >We mistakenly used a modified COLM style from another project. The camera-ready version will strictly follow the official template.
> >
> > Could you summarize the changes when using the official template (or better, if the system allows, upload a reformatted copy)? I appreciate that a camera-ready version can be reformatted, however my concern here is that with presented formatting (which appears to have smaller font and margins), a non-trivial amount of content will not fit within the allotted space for the main paper. and therefor the submitted and final papers will have significant changes.

---

> > > ### Author Response · Authors · 2025-06-04
> > > **The main difference was in margin settings, allowing more content in our initial submission. After reformatting to the official COLM template, we exceeded the limit by a few paragraphs. To resolve this, we will move Table 3 and Figure 3 to the appendix and describe them briefly in the text, ensuring compliance with COLM guidelines while retaining all essential content.**
> > >
> > > Thank you for your thoughtful comments.
> > > The primary difference between the template we initially used and the official COLM template lies in the margin settings, which inadvertently allowed us to fit more content into the submission. Upon reformatting to the correct COLM style, we found that we exceed the page limit by a few paragraphs. To address this in the camera-ready version, we plan to move Table 3 and Figure 3 to the appendix and incorporate concise textual descriptions of their contents within the main body. This adjustment will ensure full compliance with the COLM formatting guidelines while preserving the completeness of the paper’s content.

---

> > > ### Author Response · Authors · 2025-06-09
> > > **Thanks again for your thoughtful feedback. You’re right that our submitted version used slightly narrower margins, allowing in more content. We’ll fix this by moving Table 3 and Figure 3 to the appendix, summarizing them in text, and tightening phrasing. These changes keep the paper within limits without losing substance. We hope the improved clarity and completeness might warrant a score reconsideration.**
> > >
> > > We sincerely appreciate your continued engagement and thoughtful feedback.
> > >
> > > *Formatting Clarification*: You're right in noting the formatting discrepancy. The version we originally submitted used a template with slightly narrower margins, unintentionally allowing us to fit a little more content than the official COLM style permits.
> > > To resolve this, we plan to:
> > > * Move Table 3 and Figure 3 to the appendix
> > > * Condense their core insights into the main body through brief textual summaries
> > > * Tighten phrasing in a few places to regain space without affecting clarity or completeness.
> > > These adjustments will enable the final version to fit within the COLM limits without requiring substantive content changes, thereby preserving the core contributions and empirical results.
> > >
> > > Given your helpful suggestions and the clarifications we’ve now incorporated, especially around dataset complexity, model evaluation, legal realism, and expert feedback, we hope the improved clarity and completeness might warrant a reconsideration of your score.
> > >
> > > We are aware that June 10th is the final day for discussion, and we appreciate the constructive feedback you have provided throughout.

---

### Official Review · Reviewer_PPMy · 2025-05-12

**Rating:** 7
**Confidence:** 4
**Ethics Flag:** 1

**Summary:**

This work describes LawFlow, a dataset aimed at supporting LLM-based systems in the task of defining  complete end-to-end legal workflows, in contrast to current approaches that typically focus on different sub-tasks.  The framework is based on real workflows extracted by trained law students that cover from initial intake to operating agreement drafting, and are decomposed into modular subtasks (client elicitation, document review, iterative editing, ...).

The paper describes the dataset in detail and shows a comparison between LLM and human generated legal workflows, analysing the differences and detecting improvement points.

The paper is well written and motivated and addresses an interesting topic. AI applications in the legal domain have gained enormous popularity in the last decade. However, there are still gaps and this work comes to fill one of them. Particularly, it treats the legal process not atomically or focusing on the outcomes only, but as a whole (what they call "chain-of-decisions"). This study has served to highlight the limitations of LLM-based approaches, paving the way to future approaches that will add more flexibility and revisioning capabilities.

The workflow preparation and annotation process is described in detail, as well as the comparative (human vs LLMs) analysis.

The paper describes still exploratory, non-conclusive, work, but progresses in the right direction and illustrates interesting findings in automatic workflow construction.

**Reasons To Accept:**

* The paper is well written and motivated and addresses an interesting topic.
* The workflow preparation and annotation process is described in detail, as well as the comparative (human vs LLMs) analysis.

**Reasons To Reject:**

* The paper describes still exploratory, non-conclusive, work
* More technical details could be given about the LLM-based workflow preparation+ execution. For instance, they mention using two reasoning models, but not how they are combined.
* I miss a reference to the degree of competence of the involved law students. It can make a huge difference if they are undergraduate, master students, or PhD students. I also miss the number of students involved in the dataset preparation.
* The study grounds on workflows for "operating agreement" preparation. I miss a discussion on how to generalise it to other type of legal activities.

---

> ### Author Response · Authors · 2025-06-01
> **Our work provides the first empirical grounding for end-to-end legal workflows, revealing opportunities for better legal AI. We compare GPT-O1 and Deepseek-R1 using parallel execution graphs. Human workflows, authored by four advanced law students, reflect domain-agnostic patterns like iteration, contextual reasoning, and client focus, contrasting with the rigid, exhaustive structures typical of LLM-generated workflows.**
>
> We thank the reviewer for their thoughtful suggestions and feedback, which will help improve the clarity of our paper.
> * While our work is indeed exploratory, it provides the _first_ empirical grounding for end-to-end legal workflows and points to design opportunities for more effective legal AI support.
> * We will revise Section A.3.2 to provide a clear explanation of the LLM-based workflow generation. Specifically, both GPT-O1 and Deepseek-R1 are independently prompted with identical scenario contexts to simulate legal reasoning. Their outputs are not combined; instead, they are used to build parallel LLM Execution Graphs, which allow us to compare how different models navigate the same task and where their reasoning diverges.
> * Regarding participants, we recruited four high-performing third-year law students with experience in business law. One student, the acting student director of the law school’s business law clinic, assumed a leadership role in the simulations. These students are also co-authors on this paper.
> * Although LawFlow focuses on agreement writing for small business formation, its insights are applicable across various legal domains. The core human legal reasoning patterns it captures, such as iterative refinement, contextual adaptation, and client-centered decision-making, are essential to most non-trivial legal tasks. Real-world legal reasoning is rarely linear; it is chaotic and context-dependent, requiring flexible, modular workflows. LawFlow highlights these traits, contrasting them with the more rigid, exhaustive patterns typical of LLM-generated workflows.

---

> ### Comment · Reviewer_PPMy · 2025-06-03
>
> I thank the authors for their clarifications. I keep my scores.

---

### Decision · Program_Chairs · 2025-07-08

**Decision:**

Accept

**Comment:**

This work provides a useful evaluation framework, drawn from interactions with real law students to more realistically evaluate legal workflows. While still exploratory, reviewers agreed that the paper was a useful contribution to the community and recommended acceptance.